# Substantially reducing global PM$_{2.5}$-related deaths under SDG3.9 requires better air pollution control and healthcare

Huanbi Yue [1,2,3], Chunyang He [1,3,4,5] ✉, Qingxu Huang [1,3], Da Zhang [6] ✉, Peijun Shi [1,3,4,5], Enayat A. Moallemi [7], Fangjin Xu[1,3,8], Yang Yang [2,9], Xin Qi [10], Qun Ma [11] & Brett A. Bryan [12]

The United Nations' Sustainable Development Goal (SDG) 3.9 calls for a substantial reduction in deaths attributable to PM$_{2.5}$ pollution (DAPP). However, DAPP projections vary greatly and the likelihood of meeting SDG3.9 depends on complex interactions among environmental, socio-economic, and healthcare parameters. We project potential future trends in global DAPP considering the joint effects of each driver (PM$_{2.5}$ concentration, death rate of diseases, population size, and age structure) and assess the likelihood of achieving SDG3.9 under the Shared Socioeconomic Pathways (SSPs) as quantified by the Scenario Model Intercomparison Project (ScenarioMIP) framework with simulated PM$_{2.5}$ concentrations from 11 models. We find that a substantial reduction in DAPP would not be achieved under all but the most optimistic scenario settings. Even the development aligned with the Sustainability scenario (SSP1-2.6), in which DAPP was reduced by 19%, still falls just short of achieving a substantial (≥20%) reduction by 2030. Meeting SDG3.9 calls for additional efforts in air pollution control and healthcare to more aggressively reduce DAPP.

PM$_{2.5}$ pollution—particulate matter smaller than 2.5 μm in diameter suspended in the air—is one of the largest environmental risk factors for public health and has been implicated in various respiratory and cardiovascular diseases[1,2]. Deaths caused by exposure to ambient PM$_{2.5}$ pollution, termed deaths attributable to PM$_{2.5}$ pollution (DAPP)[3,4], numbered more than 4 million in 2019 worldwide[5], more than double the total reported deaths (~2 million) from COVID-19 in 2020[6]. Against this backdrop, the United Nations' Sustainable

Development Goal (SDG) 3.9 obliges signatories to "by 2030, substantially reduce the number of deaths and illnesses from hazardous chemicals and air, water and soil pollution and contamination[1,7]". Within the total deaths related to air pollution (generally considered PM$_{2.5}$ and ozone), DAPP accounts for more than 95%[8,9]. Hence, understanding future trends in DAPP and the relative influence of key drivers is of great importance for environmental policy and public health management.

[1]Key Laboratory of Environmental Change and Natural Disasters of Chinese Ministry of Education, Beijing Normal University, Beijing, China. [2]School of International Affairs and Public Administration, Ocean University of China, Qingdao, China. [3]State Key Laboratory of Earth Surface Processes and Resource Ecology, Beijing Normal University, Beijing, China. [4]Academy of Disaster Reduction and Emergency Management, Ministry of Emergency Management & Ministry of Education, Beijing Normal University, Beijing, China. [5]Academy of Plateau Science and Sustainability, People's Government of Qinghai Province & Beijing Normal University, Xining, China. [6]College of Geography and Ocean Sciences, Yanbian University, Yanji, China. [7]Commonwealth Scientific and Industrial Research Organization (CSIRO), Melbourne, Victoria, Australia. [8]College of Urban and Environmental Sciences, Peking University, Beijing, China. [9]Institute of Marine Development, Ocean University of China, Qingdao, China. [10]Frontiers Science Center for Deep Ocean Multispheres and Earth System (FDOMES), Ocean University of China, Qingdao, China. [11]School of Environmental and Geographical Sciences, Shanghai Normal University, Shanghai, China. [12]School of Life and Environmental Sciences, Deakin University, Melbourne, Victoria, Australia. ✉e-mail: hcy@bnu.edu.cn; zhangda@ybu.edu.cn

DAPP is jointly driven by interactions between atmospheric $PM_{2.5}$ concentration, population size, population age structure, and the healthcare (presented as death rate of diseases), which in turn, are affected by socioeconomic development and climate change (Fig. 1). Specifically, socioeconomic development directly influences the total population, age structure, and healthcare standards[10,11], and has an indirect effect on $PM_{2.5}$ concentration via the emission of airborne pollutants from the manufacturing, energy, and transport sectors[12,13]. Climate change can also affect $PM_{2.5}$ concentration by influencing wind, precipitation, and other parameters which determine the transport and evolution of pollutants[14,15]. Socioeconomic development and climate change are also complexly interrelated[16].

While a few studies have projected future DAPP, estimates vary widely, with some studies showing opposite trends (Table S1). For instance, Lelieveld et al.[17] and Rafaj et al.[18] projected future increases in DAPP, while West et al.[19] and Silva et al.[15] projected declines. These studies have mainly focused on the effects of changes in $PM_{2.5}$ concentration on DAPP, while other influential factors have either been held constant or estimated via projections from bespoke scenario frameworks. For example, some studies have assumed a constant death rate of diseases or age structure to isolate the effect of future air quality change on DAPP[17,20], while other studies used projected $PM_{2.5}$ concentration based on climate scenarios but assumed trends in demographic factors or death rate of diseases from ad hoc projections[15,21]. This lack of internal consistency in scenario assumptions cannot reliably capture relative effects of or the interactions between driving factors[15,19,22] and can lead to inconsistent and even contrasting results. Furthermore, Yang et al.[23] projected DAPP under an integrated scenario framework which simulated $PM_{2.5}$ concentration from a single atmospheric model (GFDL-ESM4.1), hence the results are limited in capturing the range of variation in $PM_{2.5}$ concentration and the corresponding DAPP under climate change[15]. With less than 7 years to go to achieve the SDGs, a comprehensive projection of global DAPP at high spatial resolution, using multiple models and considering each driving factor and the inherent uncertainty in a coherent and internally consistent way is both essential and urgent.

In this study, we aimed to produce an integrated, comprehensive, and coherent suite of projections of future DAPP by using future $PM_{2.5}$ concentration based on 11 available models (see details in Methods) and CMIP6 (Coupled Model Intercomparison Project Phase 6) climate change projections under the latest Scenario Model Intercomparison Project (ScenarioMIP) framework (Table 1). We considered four integrated scenarios represented as SSP $x$–$y$ (i.e., Sustainability (SSP1-2.6); Middle of the Road (SSP2-4.5); Regional Rivalry (SSP3-7.0), and Fossil-fueled Development (SSP5-8.5)) in exploring the potential to achieve SDG3.9 by substantially reducing DAPP for 154 countries worldwide (Figure S1), where $x$ represents societal conditions described by Shared

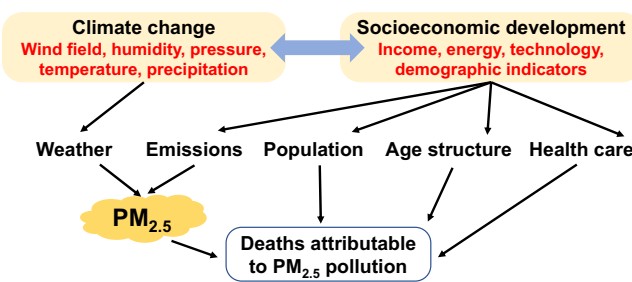

**Fig. 1 | The impact of socioeconomic development and climate change on deaths attributable to $PM_{2.5}$ pollution.** Socioeconomic development pathways can lead to changes in population, age structure, death rate of diseases, and $PM_{2.5}$ concentration. Note that $PM_{2.5}$ concentration is jointly determined by emissions and climatic conditions which are both a product of complex interactions between socioeconomic development and climate change.

**Table 1 | General descriptions of selected scenarios**

| Scenario | Sustainability (SSP1-2.6) | Middle of the Road (SSP2-4.5) | Regional Rivalry (SSP3-7.0) | Fossil-fueled Development (SSP5-8.5) |
|---|---|---|---|---|
| Air pollution control | Strong air pollution control with widespread adoption of current best available technology and overall enforcement of environmental laws. Pollutant targets are substantially more ambitious than current levels. | Medium air pollution control with implementation of advanced technologies over current levels, combined with some 'catch-up' for low- and medium-income countries in policy efficacy and more ambitious concentration targets. | Weak air pollution control driven by limited technological improvement and environmental awareness. The targets are less ambitious than SSP2-4.5. | Same as SSP1-2.6. |
| Energy demand | Low energy demand due to technological development, lifestyle changes and energy efficiency improvements. The share of fossil fuel is replaced by clean energy. | Medium-high energy demand with energy intensity improvement as trend. No remarkable shifts in technology and energy mix. | Medium energy demand caused by slow technological development, material intensive lifestyles and little environmental awareness. Traditional fossil fuel use remains important. | High energy demand strongly coupled to economic activities. Mitigation technologies developed rapidly in the fossil fuel sector with a high social acceptance. |
| Population | Low world population associated with lower fertility and higher life expectancy. | Medium population growth following the historical trend. | High population associated with higher fertility and lower life expectancy. | Same as SSP1-2.6. |
| Economic development | Intermediate economic growth due to quicker convergence. | Medium economic growth following the historical trend. | Very low growth following weak international co-operation and trade. | High economic growth with focused on "conventional" economic development |
| Radiative forcing level | 2.6 W/m² at 2100. Corresponding with the RCP2.6 pathway. | 4.5 W/m² at 2100. Corresponding with the RCP4.5 pathway. | 7.0 W/m² at 2100. Corresponding with the RCP7.0 pathway. | 8.5 W/m² at 2100. Corresponding with the RCP8.5 pathway. |

SSP refers to the Shared Socioeconomic Pathway; RCP is the Representation Concentration Pathway. Adapted from O'Neill et al.[62], Rao et al.[62], Riahi et al.[66], and Riahi et al.[64].

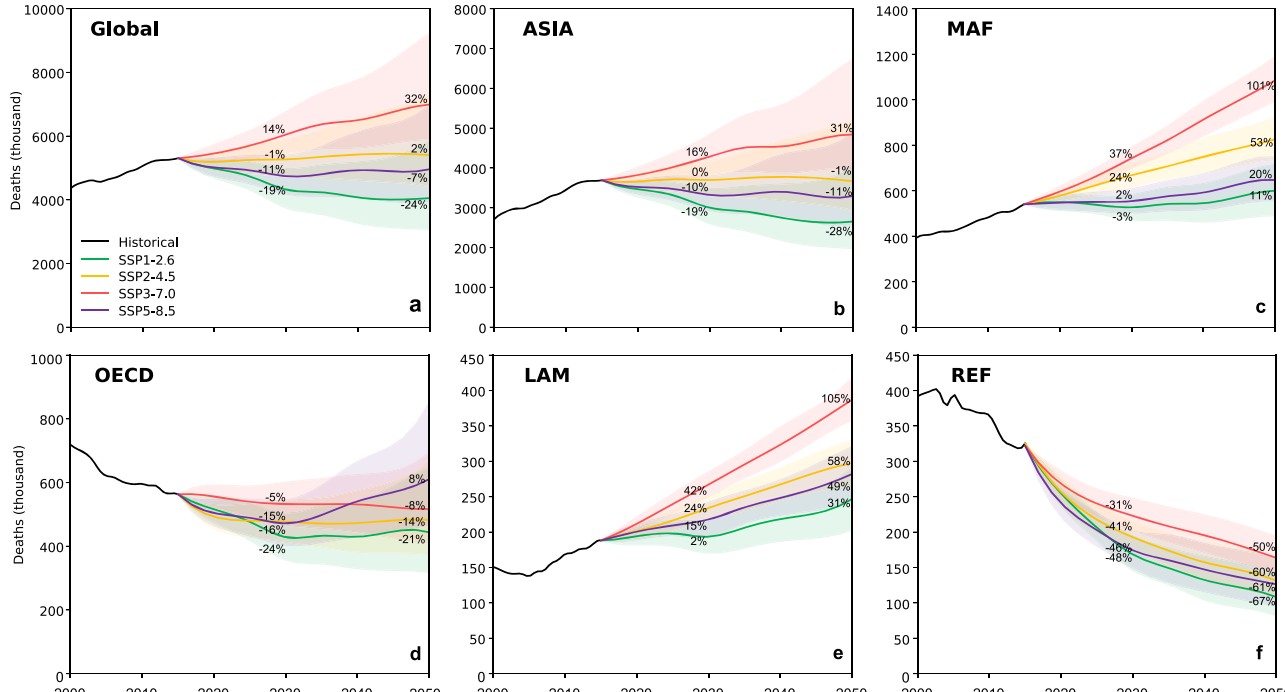

**Fig. 2 | Historical changes in deaths attributable to PM₂.₅ pollution (DAPP) and projected attainment of SDG3.9 to 2050. a** Changes in DAPP at the global scale, **b–f** regional scale. Solid lines represent the average estimates and shading indicates the 95% confidence interval derived from uncertainty in future PM₂.₅ concentration (derived from 11 climate and earth system models) and the death rate of diseases (derived from the statistic model). See details in Supplementary Notes 2-4. The abbreviation is defined as ASIA Asian with the exception of the Middle East, Japan, and Former Soviet Union states. MAF Middle East and Africa. LAM Latin America and the Caribbean. OECD Organization for Economic Co-operation and Development and new European Union and candidates. REF reforming economies of Eastern Europe and the former Soviet Union.

Socioeconomic Pathway (SSP) and *y* represents the degree of climate forcing under the Representative Concentration Pathway that is consistent with SSP emissions[24]. First, we projected global DAPP from 2015 (adoption of the UN Agenda 2030) to 2050 by combining an epidemiological model with all driving factors under the ScenarioMIP framework (Figure S2) and assessed the attainment of SDG3.9 at three levels of ambition (10, 20, and 30% reduction compared to 2015 levels). We then identified the relative contributions of changes in PM₂.₅ concentration, population size, age structure, and death rate of diseases on DAPP using the decomposition method (Figure S18). Lastly, we explored alternative pathways to meet SDG3.9 by leveraging additional effort in air pollution control (20% lower PM₂.₅ concentration) and healthcare improvement (20% lower death rate of diseases) (see details in Methods). The results are crucial for informing national-level investment and policy for teaming climate change mitigation, air pollution control, and healthcare to substantially reduce global DAPP.

## Results

### Trends in DAPP under different scenarios and the attainment of SDG3.9

The multi-model average results indicate that global DAPP was substantially reduced only under those scenarios with the most ambitious assumptions around continuing growth and aging in the global population and declines in death rates of diseases (Figure S15). Thus, achieving SDG3.9 remains a great challenge. The SSP1-2.6 scenario saw the largest decrease, with average DAPP projected to almost meet the moderate target (i.e., a 20% reduction compared to 2015 levels) by 2030 (−19%) and exceed the target by 2050 (−24%). Under all other scenarios, the moderate target of SDG3.9 was not achieved. Average DAPP also declined (−11% by 2030 and −7% by 2050) under the SSP5-8.5 scenario, but remained stable under the SSP2-4.5 scenario (−1% by 2030 and +2% by 2050). In the worst case, average DAPP grew 14% by

2030 and 32% by 2050 under SSP3-7.0 (Fig. 2a). Among different age groups, the share of DAPP for older people (65 +) accounted for almost 65% in 2015 and rose to 70% under all scenarios by 2030 because older adults have a higher baseline death rate and are more vulnerable to almost all types of health risk (Figure S5).

Middle East and Africa (MAF) and Latin America and the Caribbean (LAM) were hotspots of future growth in DAPP, increasing between 11% (SSP1-2.6) and 101% (SSP3-7.0) by 2030, and between 13% (SSP1-2.6) and 105% (SSP3-7.0) by 2050, respectively. SDG3.9 achievement in these regions is not expected in the future (Fig. 2c, e). In contrast, DAPP in the member states of the Organization for Economic Co-operation and Development and new European Union and candidates (OECD) and the reforming economies of Eastern Europe and the Former Soviet Union (REF) tended to decrease under most scenarios. The OECD met the moderate SDG3.9 target under SSP1-2.6 by 2030 and 2050 while REF even achieved the ambitious target (30% reduction) (Fig. 2d, f).

Future trends in PM₂.₅ concentrations among different CMIP6 general circulation models differed in both magnitude and sign because of their differences in natural emissions, chemical mechanisms, and processes. The multi-model average DAPP was estimated based on all available models to provide a general trend (see details in Methods). To further encompass the uncertainties among the 11 models, we also calculated the model-specific trend in DAPP (Figure S9) and found that most models yielded results similar to the average estimates (calculated based on all available models). Although some models (e.g., MIROC-ES2L, INM-CM5-0) displayed a steeper decline in DAPP, the moderate SDG3.9 target was not achieved by 2030 for any scenario except SSP1-2.6 (Figure S4; Table 2). Considering some models were scenario-specific, to ensure consistency, we also presented the average results calculated based on the eight models with estimates for all scenarios, which was highly in line with average results from all available models.

Achieving the moderate SDG3.9 target was also a challenge at national scale. According to the multi-model average DAPP, more than two-thirds of the world's nations (107/154) did not meet the moderate target by 2030 under any scenario (Fig. 3), including over 80% of countries in LAM and MAF, and more than 70% of countries in ASIA (Table S2). Even by 2050, although the number of countries achieving the moderate SDG3.9 target varied slightly over time, substantially reducing DAPP remained a challenge for most countries (Fig. 3). When loosening the definition of a "substantial reduction" to a 10% reduction (i.e., the weak target), the challenge to meet SDG3.9 remained with over 50% of nations (87/154) failing to meet SDG3.9 by 2030 under any scenario, and when tightening (30%) the target (i.e., the ambitious target), 80% of nations (127/154) failed (Fig. 3).

While considering the variation among different models, the finding many countries cannot achieve SDG3.9 under any scenario was robust. Even when considering the relatively loose standard for SDG3.9 (i.e., a 10% reduction) and the model with the most aggressive $PM_{2.5}$

pollution reduction (MIROC-ES2L), more than one-third of nations failed to achieve a substantial reduction in DAPP. This proportion rose to 55% and 69% when tightening the target to moderate (i.e., 20%) and ambitious (i.e., 30%).

Within regions, substantial spatial heterogeneity occurred (Fig. 4), with global concentrations in DAPP coinciding with the major global urban centers where dense populations co-exist with the major sources of $PM_{2.5}$ pollution (i.e., traffic networks, power generation, heavy industry). For example, under the Middle of the Road scenario (SSP2-4.5), northern India had the largest growth in DAPP which included the megacities of Delhi and Kolkata. Similarly, the largest city in South America−Sao Paulo in Brazil−also saw a large growth in DAPP.

### Effects of individual factors on the attainment of SDG3.9

Trends of DAPP can be disentangled into the net effects of changes in $PM_{2.5}$ concentration which reflects the risk factor itself, and population, age structure, and death rate of diseases which alter the size and

**Table 2 | Changes in deaths attributable to $PM_{2.5}$ pollution (DAPP) under different CMIP6 general circulation (climate) models**

| Change in DAPP relative to 2015 (%) | Scenarios | Average | Average-8 models | GFDL-ESM4 | GISS-E2-1-G | INM-CM4-8 | INM-CM5-0 | MIROC-ES2L | MRI-ESM2-0 | NorESM2-LM | NorESM2-MM | GFDL-CM4 | BCC-ESM1 | CNRM-ESM2-1 |
|---|---|---|---|---|---|---|---|---|---|---|---|---|---|---|
| By 2030 | SSP1-2.6 | −18.9 | −18.7 | −20.0 | −16.4 | −19.9 | −20.8 | −22.5 | −18.3 | −16.0 | −17.1 | N/A | N/A | N/A |
| | SSP2-4.5 | -0.9 | −0.7 | −2.5 | −0.6 | −1.4 | −2.8 | −2.5 | 0.1 | 0.9 | 1.4 | −0.4 | N/A | N/A |
| | SSP3-7.0 | 13.8 | 13.4 | 15.2 | 16.1 | 10.8 | 11.1 | 14.7 | 13.7 | 14.8 | 12.6 | N/A | 14.3 | 14.2 |
| | SSP5-8.5 | −11.0 | −10.8 | −11.0 | −12.6 | −13.1 | −13.9 | −14.0 | −10.5 | −5.4 | −6.2 | −12.6 | N/A | N/A |
| By 2050 | SSP1-2.6 | −24.6 | −24.9 | −22.8 | −19.1 | −36.3 | −38.7 | −28.4 | −20.8 | −15.9 | −15.0 | N/A | N/A | N/A |
| | SSP2-4.5 | 1.2 | 1.1 | 0.4 | 3.6 | −6.1 | −6.4 | −2.6 | 7.1 | 6.3 | 6.0 | 2.7 | N/A | N/A |
| | SSP3-7.0 | 31.4 | 30.6 | 35.6 | 32.5 | 28.3 | 26.9 | 30.8 | 32.4 | 32.6 | 30.7 | N/A | 31.8 | 32.0 |
| | SSP5-8.5 | −7.3 | −7.2 | −5.3 | −5.6 | −15.5 | −16.2 | −12.5 | −4.4 | 4.9 | −0.9 | −10.4 | N/A | N/A |

The DAPP was estimated based on ensemble average $PM_{2.5}$ concentration from different models.

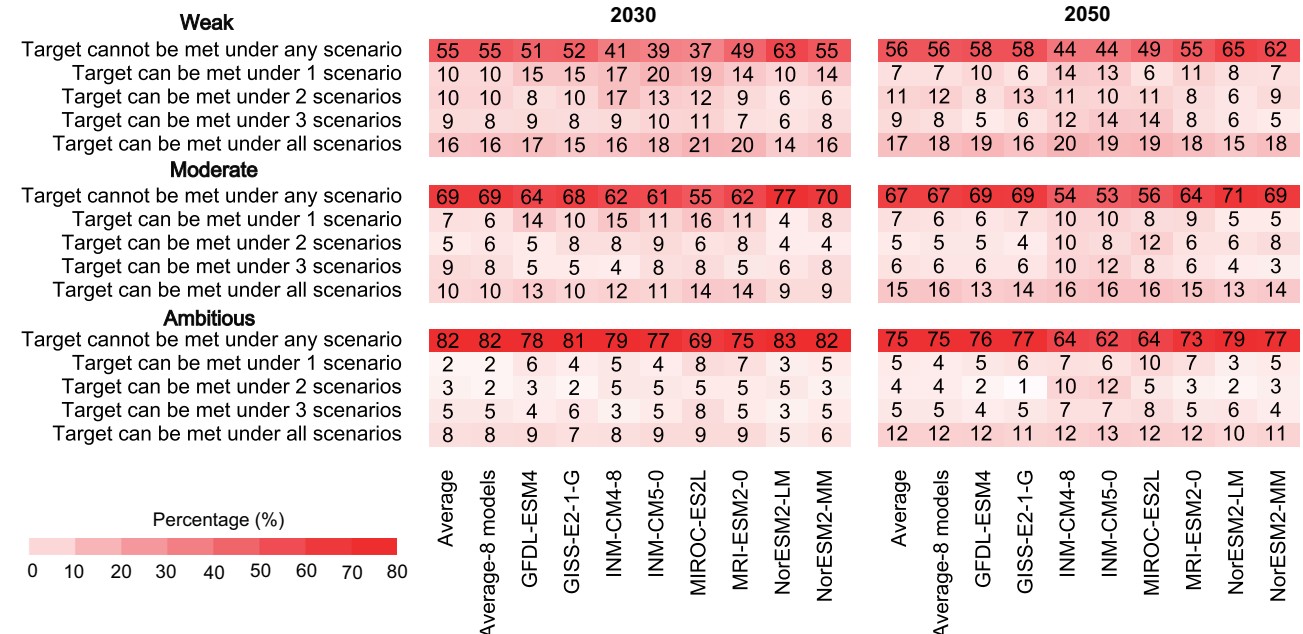

**Fig. 3 | Attainment of SDG3.9 by 2030 and 2050 for the 154 nations.** The colors indicate percentages of countries that can meet SDG3.9 under 0, 1, 3, 2, and 4 possible scenarios assessed with, weak, moderate, and ambitious settings for SDG3.9 represented as 10%, 20%, and 30% reduction in DAPP relative to 2015, respectively. Only 8 models that were available for all the 4 scenarios were included in the model-specific analysis.

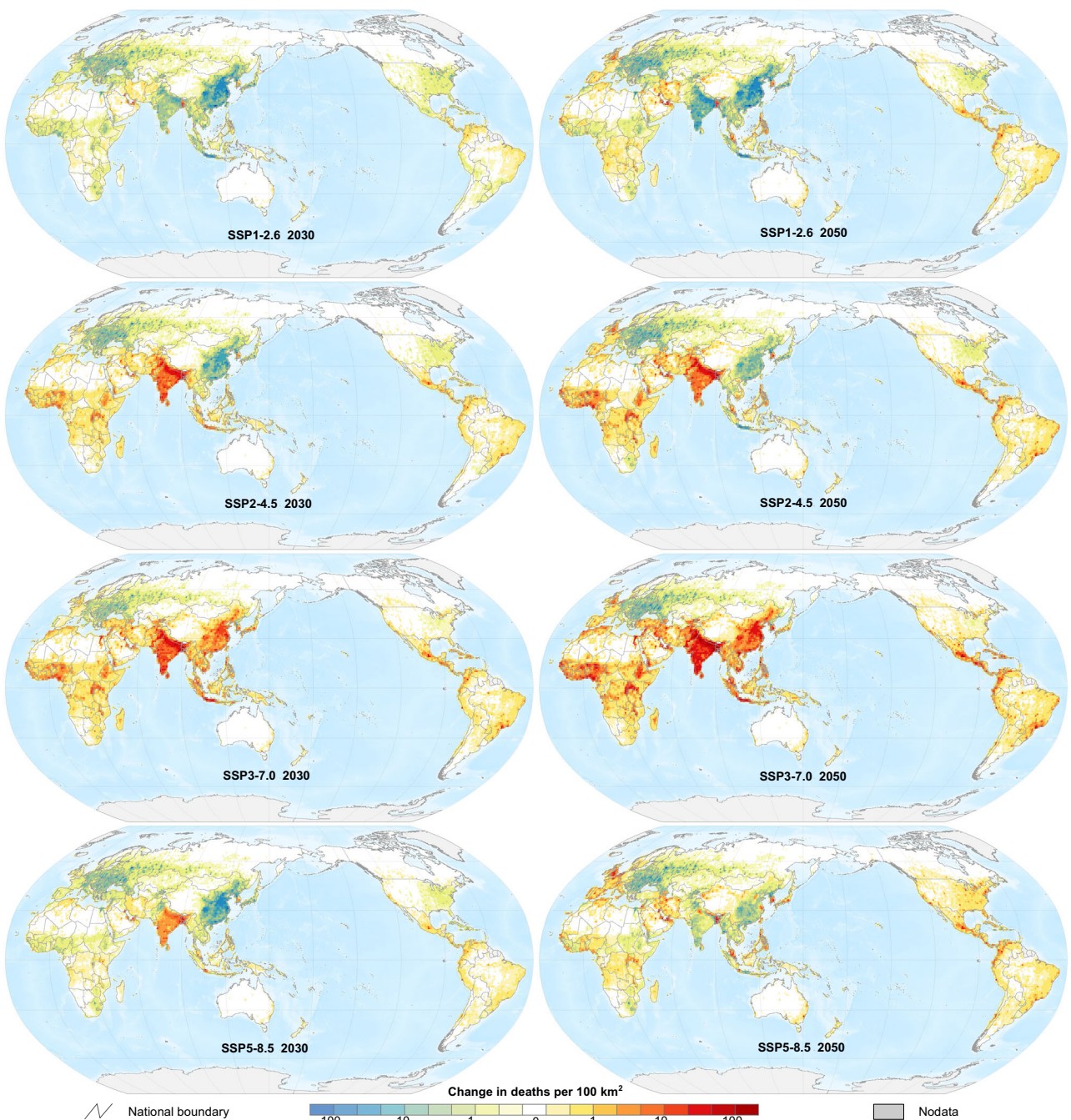

**Fig. 4 | Spatially-explicit changes in deaths attributable to PM$_{2.5}$ pollution relative to 2015 under different scenarios.** Results estimated based on multimodel average PM$_{2.5}$ concentration. The base map is made by authors with national boundary data from the National Platform for Common Geospatial Information Services of China[75] and water depth data from ETOPO dataset[76].

vulnerability via decomposition analysis (see Methods for details). Population aging was the dominant factor under all scenarios in driving growth in DAPP, causing an increase of 34% (SSP3-7.0) to 41% (SSP5-8.5) by 2030. Conversely, healthcare improvement was the strongest factor driving down DAPP, leading to a decline of 36% (SSP3-7.0) to 44% (SSP5-8.5). As socioeconomic development reduced the death rate of diseases and exacerbated population aging simultaneously, these two drivers tended to offset each other, resulting in a relatively minor net decrease of around 1 to 3% (Fig. 5). This counteracting effect also influenced the age distribution of DAPP. The share of DAPP amongst older people (65 + ) increased from almost 65% in 2015 to around 70% under all scenarios by 2030 due to the increase in older adults and better healthcare (Figure S5).

Population growth led to a minor increase in DAPP of around 6% (SSP1-2.6) to 14% (SSP3-7.0).

Compared with other driving factors, the effect of PM$_{2.5}$ concentration was more variable across scenarios. For instance, DAPP in 2030 was projected to decrease by 24% relative to 2015 because of the decline in PM$_{2.5}$ concentration alone under SSP1-2.6, led by the effort in climate mitigation and strong air pollution control. Similarly, air quality improvement was responsible for a decline of 15% and 8% in DAPP under SSP5-8.5 (strong air pollution control and high energy demand) and SSP2-4.5 (moderate air pollution control and moderate energy demand), respectively. Under the worst scenario for air pollution (SSP3-7.0) which assumes a weak air pollution control with slow socioeconomic development, DAPP was projected to increase by 2%

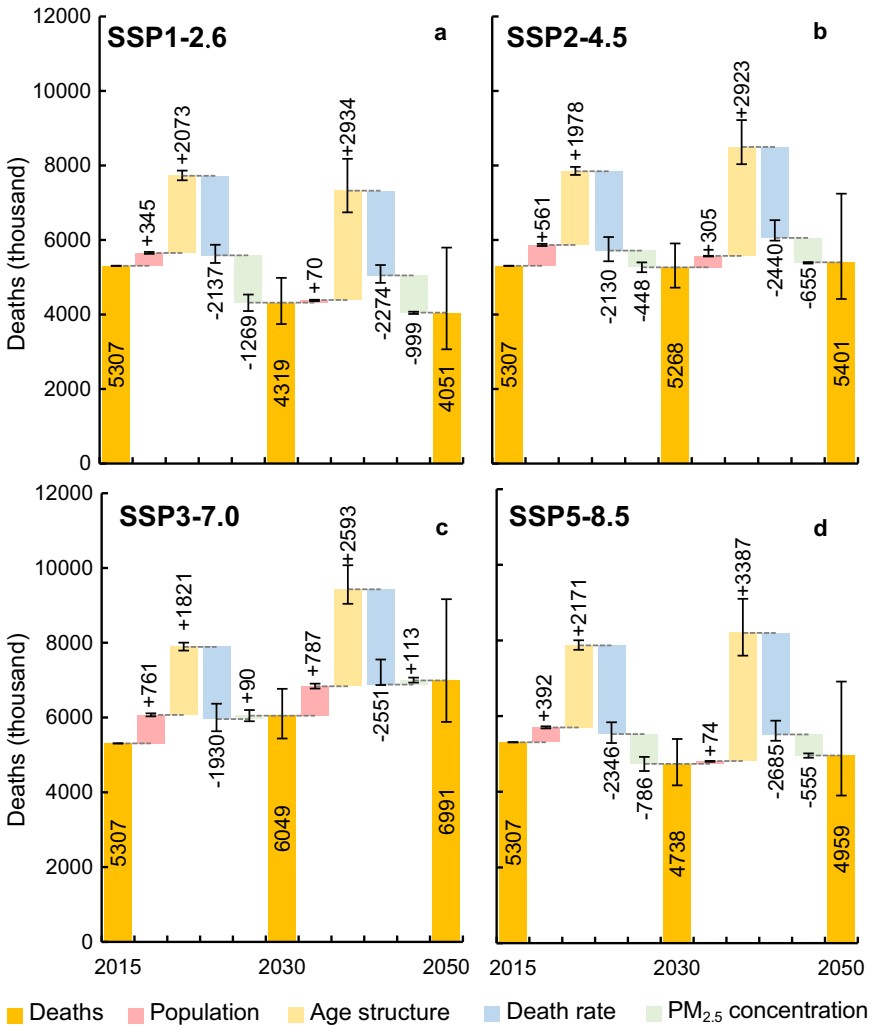

**Fig. 5 | Contributions of different factors to changes in deaths attributable to PM$_{2.5}$ pollution. a–d** the cumulative effect of four factors: population, age structure, death rate, and PM$_{2.5}$ concentration, under SSP1-2.6, SSP2-4.5, SSP3-7.0, and SSP5-8.5, respectively. Data represents the mean value and error bars represent the projections based on the 95% confidence intervals of future PM$_{2.5}$ concentration (derived from 11 climate and earth system models) and the death rate of diseases (derived from a statistical model). See details in Supplementary Notes 2-4.

due to a 12% increase in PM$_{2.5}$ concentration relative to 2015 (Fig. 5; Figure S15). As the relationship between PM$_{2.5}$ concentration and health effects represented by exposure-response curves is non-linear, DAPP tended to show a steeper increase at lower concentrations and leveled off at higher concentrations (Figure S3). Hence, a lower level of PM$_{2.5}$ concentration in the future would substantially reduce DAPP. For example, under SSP1-2.6 a decrease in PM$_{2.5}$ concentration of 31% led to a 24% decrease in DAPP from 2015 to 2030; whereas under SSP5-8.5, a growth of 12% in PM$_{2.5}$ concentration led to a 2% increase in DAPP (Fig. 5).

Similar patterns in driving factors occurred beyond 2030 at the global scale. Aging and healthcare improvement remained the dominant driving factors affecting DAPP and the attainment of SDG3.9 while the effect of PM$_{2.5}$ concentration varied among scenarios. The effect of population growth between 2030 and 2050 was smaller relative to that from 2015 to 2030 because population growth was slower.

Detailed region- and country-specific decomposition were also conducted (Figure S19-20). For most countries, changes in aging and healthcare were the main contributors to DAPP but the interaction of other driving factors also had an influence. For instance, DAPP in China declined over time and met moderate or ambitious SDG3.9 targets under all scenarios except SSP3-7.0 because of low population growth and decreasing PM$_{2.5}$ concentration. Conversely, DAPP in India tended

to increase in most scenarios driven by population growth and relatively modest and slow air quality improvement.

## Meeting SDG3.9 via additional pollution control and healthcare advances

Amongst the four driving factors that shape DAPP, PM$_{2.5}$ concentration and death rate of diseases are more readily modified via policy measures such as incentives, regulation, and investment in technology research, development, and implementation. Hence, we conducted a sensitivity analysis to explore the ability to meet SDG3.9 via additional efforts in air pollution control and healthcare (see details in Methods).

The results indicate that advances in air pollution control and healthcare can make important contributions towards achieving a substantial reduction in DAPP. When PM$_{2.5}$ concentration was set at 20% lower (i.e., moderate SDG3.9 target) than the projected value under different scenarios, a substantial reduction (-29%) in global DAPP was achieved under SSP1-2.6 by 2030. Under SSP5-8.5, a decrease in DAPP of 21% was achieved. When the death rate of diseases was set at 20% lower than the projected value, the moderate SDG3.9 target was met under most scenarios except SSP3-7.0. With both measures in place (i.e., a 20% lower PM$_{2.5}$ concentration and a 20% lower death rate of diseases), large reductions in DAPP were achieved (Fig. 6a).

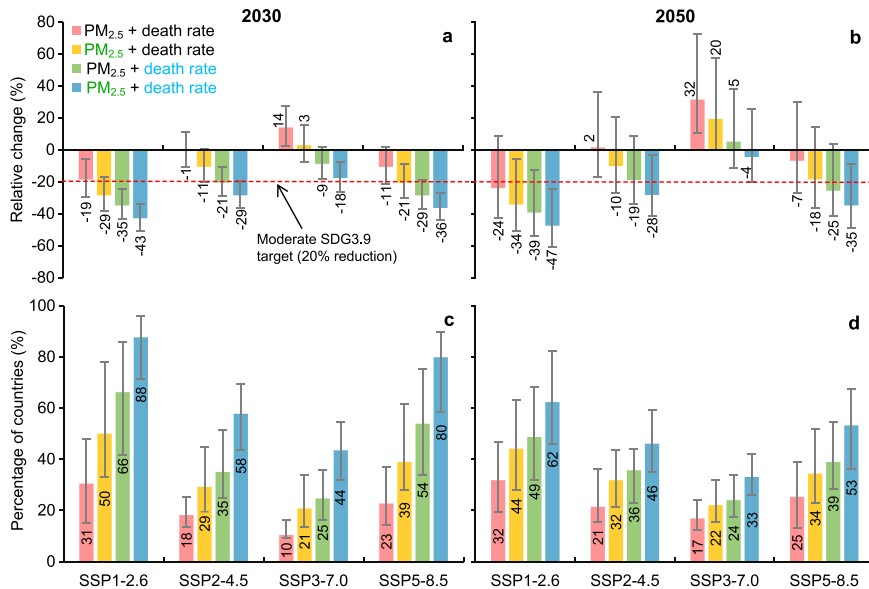

**Fig. 6 | The potential effect of additional improvement in air pollution control and healthcare on the attainment of the moderate SDG3.9 target. a**, **b** Relative change in deaths attributable to PM$_{2.5}$ pollution from 2015 to 2030 and 2050. **c**, **d** Percentage of countries meeting moderate SDG3.9 by 2030 and 2050 (calculated based on 154 countries). Colors in the legends indicate a 20% lower PM$_{2.5}$ concentration (green) and death rate of diseases (blue) relative to the projected value because of additional improvements in air pollution control and healthcare.

The data represent the mean value and error bars represent 95% confidence intervals, calculated based on the upper and lower intervals of future PM$_{2.5}$ concentration (derived from 11 climate and earth system models) and the death rate of diseases (derived from statistic model). See details in Supplementary Notes 3-4. A similar reduction was also adopted to the upper and lower intervals when considering the additional improvement.

Thus, the challenge to substantially reduce DAPP for countries requires substantial advances in both healthcare and air pollution control which can boost the reduction of DAPP for countries that did not meet SDG3.9 under the original ScenarioMIP scenarios. By combining PM$_{2.5}$ concentrations and the death rate of diseases at 20% lower than the projected values, the number of countries that met SDG3.9 more than doubled relative to the original projections, and almost 90% of the world's nations can achieve the moderate SDG3.9 target under SSP1-2.6 by 2030 (Fig. 6c). Details of the potential pathways for each country to meet the moderate SDG3.9 target by considering additional improvement in healthcare and air pollution control can be found in Figure S21.

## Discussion

Several studies have assessed the trends in DAPP including dynamic PM$_{2.5}$ concentration levels, population structure and aging, and death rates of diseases based on various scenarios (including SSP-RCP scenarios) (Table S1). However, a lack of internally consistent, comprehensive scenario analyses has led to wide variations in future projections of DAPP which differ not only in magnitude but also in sign, potentially resulting in divergent interpretations and policy responses. As a response, a major contribution of this study is providing a comprehensive long-term projection of DAPP with all 11 available simulations from CMIP6 under the latest ScenarioMIP framework, which provides ensembles of integrated, internally consistent estimates for the drivers of DAPP (population, age structure, death rate of diseases, and air quality)[25]. Thus, our projections used internally consistent assumptions around driving factors[15] to improve the accuracy and robustness of projections. Compared with another study[23] that did project DAPP using coherent and internally consistent assumptions for each relevant factor under the ScenarioMIP framework with PM$_{2.5}$ concentration simulated by GFDL-ESM4.1, our study also provides all available CMIP6 general circulation models. In addition, estimation based on a well-established scenario framework also makes our results comparable with other studies under the SSPs and

CMIP6 framework and estimates the trends in DAPP under different climate change and socioeconomic pathways. This can further support integrated decision-making by jointly considering the results from multiple fields[26,27].

This work also attempted to assess potential future progress and development pathways towards SDG3.9 and does so on multiple scales: global, regional, national, and by grid cell at a spatial resolution of 1 degree. The results shed light on guiding future air pollution control policy and healthcare improvement and provide the robust projections required to support the management of health impacts caused by PM$_{2.5}$ pollution for the scientific community and stakeholders.

Achieving SDG3.9 requires additional efforts in air pollution mitigation and healthcare beyond the SSP storylines. Global DAPP will not be substantially reduced under all but the most ambitious scenarios (SSP1-2.6) because of the overwhelming effect of population aging (Fig. 3). However, there are complex interactions between drivers under different scenarios which must be considered in terms of the net overall effect on DAPP. For instance, with socioeconomic development, the death rate of diseases declines reduces population vulnerability, but population aging simultaneously increases the size of vulnerable populations[11,28]. Hence, while scenarios with high population aging (SSP1-2.6 and SSP5-8.5) counterintuitively showed a greater reduction in DAPP, this was driven by the effect of population aging being offset by improvement in healthcare in these scenarios. Besides the improvement in healthcare, change in PM$_{2.5}$ concentration is also a key driver of reducing DAPP. SSP1-2.6 had the largest reduction because of the underlying assumption of a cleaner energy mix and strong air pollution control. However, PM$_{2.5}$ concentration is also expected to reduce in SSP5-8.5. That is because of the strong air pollution control led by the adoption of the current best available technology (especially end-of-pipe control) and strong socio-economic development based on high energy demand and fossil fuel dominated energy mix. Hence, socioeconomic transformation towards a more sustainable future aligned with SSP1-2.6 with stronger air quality

control and less fossil fuel dependence is the better pathway for reducing DAPP (Fig. 5, Figure S9) as it has positive co-benefits for many other SDGs that are not assessed in this study.

In addition, the interaction between different drivers varied by region. For instance, besides population aging, population growth in MAF also played a considerable role under all scenarios because the region's population is relatively young and growing fast. While in LAM and ASIA, aging is the dominant driver of growth in DAPP (Figure S19). Different countries also exhibited heterogeneity in mechanisms driving DAPP. For instance, China is likely to achieve SDG3.9 under most scenarios because of the reversal of population growth and the declining trend of $PM_{2.5}$ pollution, as well as healthcare improvement. While for India, DAPP tended to increase under three scenarios (all bar SSP1-2.6) mainly because of the joint effect of air quality deterioration and demographic change.

With the challenge in the reduction of DAPP, meeting SDG3.9 needs strong additional policies and investment in air pollution control and healthcare. While considering the additional improvement in air quality and healthcare alone, weak and moderate SDG3.9 targets can be achieved under SSP2-4.5, and while combining both 20% lower $PM_{2.5}$ concentration and death rate of diseases, even an ambitious SDG3.9 target (i.e., a 30% reduction in DAPP) can almost be achieved. Under the most ideal settings combining SSP1-2.6, an additional 20% lower death rate of diseases, and 20% lower $PM_{2.5}$ concentration, the more ambitious target for SDG3.9 can be exceeded by 2030 with a decrease in DAPP of 43%.

ScenarioMIP provided an ensemble description of feasible futures with integrated transitions across multiple sectors including pollution control, energy structure, emissions reduction, technology innovation, and socioeconomic development (Table 1). Beyond the measures incorporated in the SSP-based narratives, extra efforts to control air pollution need to be implemented if SDG3.9 is to be achieved, such as promoting end-of-pipe devices, replacing fossil fuel with renewable electricity, and improving efficiency via technology upgrade[29-31]. An additional decrease in the death rate of disease can be achieved by measures such as strengthening the investment in the health system, building a health monitoring system for older people, and increasing medical accessibility through financial support and allied services provision[32-34].

Due to the heterogeneity in pathways to reduce air pollution and improve healthcare, countries also need different strategies to meet SDG3.9 (Figure S21). For countries like China where it is feasible to meet SDG3.9 by following the basic scenarios, future development aligned with less fossil fuel dependency and with more stringent pollution controls (i.e., SSP1-2.6) is needed. In countries like India that cannot meet SDG3.9 under any of the basic scenarios, more investment in emissions reduction and healthcare improvement are needed. Moreover, many developing countries in MAF and LAM face great challenges to meeting SDG3.9 under most scenarios even with additional improvements in healthcare and air pollution control. These countries will need more assistance from developed countries (such as high-income countries identified by the World Bank) in technology, medicine, and finance. Finally, even for the countries that meet SDG3.9 under most scenarios, efforts in air quality control and emissions abatement can be helpful to boost the well-being of residents and offset potential medical expenditure[34-37]. Cost-effectiveness and technological feasibility should also be considered. Strategies like pricing carbon and pollutant emissions using market-based policy could be a feasible way to help developing countries encourage innovation[38].

Stronger efforts in air pollution control and healthcare are also co-beneficial for achieving several other SDGs. Efforts in air pollution control are in line with mitigation and adaptation commitments in SDG13 (Climate Action)[39,40] and promote progress toward many other aspects of sustainability. Hence, comprehensive policies that team up air pollution control and public health with climate change mitigation efforts, technological innovation, and energy system overhaul could help meet multiple SDGs[41-43].

A few key limitations and uncertainties remain in our study. Firstly, in estimating DAPP, although we used the latest method from GBD 2019[5], the epidemiological model cannot differentiate the interactive effects between $PM_{2.5}$ pollution and other closely related risk factors (e.g., climate change-induced heatwaves, ozone pollution) and consideration of these effects is required in future analyses[44,45]. As our model reflected the long-term effects of $PM_{2.5}$, and estimated DAPP based on annual $PM_{2.5}$ concentration, the short-term effects associated with acute cardiovascular and respiratory diseases were not considered[46]. Further estimations of DAPP considering short-term epidemiological studies are needed in the future[47]. In addition, because of a lack of indoor $PM_{2.5}$ data, our study did not differentiate the interaction between ambient and indoor $PM_{2.5}$[48], but this limitation does not influence our major findings or conclusions.

Secondly, the choice of data sources for quantifying the driving factors and the exposure-response curve used both introduce uncertainty into DAPP projection[49]. Regarding $PM_{2.5}$ concentration, there are several potential explanations for the discrepancy between the 11 CMIP6 models, including differences in the treatment of aerosols and their components (e.g., organic aerosols and emission of biogenic volatile organic compounds) as well as the effect of climate change (i.e., temperature and precipitation) simulated by models and its impact on natural aerosol emissions[50]. To account for this variation, we presented model-specific (Figure S9) as well as multi-model average results with ranges (Figure S15-16) reflecting uncertainty in future $PM_{2.5}$ concentration. Regarding the exposure-response function, we used the Bayesian, regularized, trimmed (MR-BRT) model following GBD2019 (see details in Methods), and presented the results derived from the middle value, and the upper and lower estimates was also conducted to quantify the range of uncertainty (Figure S7-8, Figure S22). Some studies also found a potential shortcoming of the MR-BRT model in that the saturation of relative risk at high $PM_{2.5}$ tends to be more pronounced. More comprehensive sensitivity analyses considering other exposure-response functions (e.g., Global Exposure Mortality Model and Fusion model) are needed in the future to more thoroughly explore the influence of this modeling choice[51]. Considering the availability of data, projections of death rate of diseases and age structure in this study were conducted at the national level, which generalizes the heterogeneity within countries, an effect which may be critical especially in large and populous countries (e.g., United States, China, India). While we do not provide a detailed exploration of the results at the national-level here, further analyses of trends and driving mechanism for individual countries may be undertaken based on the Supplementary Information provided to support decision-making of communities and stakeholders at regional and local scales[52-54].

Thirdly, our projections were conducted under the ScenarioMIP framework, which reflected possible future socioeconomic and climate trajectories. The results should be treated as projections based on interactions among different driving factors, rather than as forecasts. Our study cannot reflect the effects caused by abrupt socio-economic, geopolitical, or environmental changes (such as war or pandemic disease)[55-57] or specific emissions control actions (such as China's Air Pollution Control Policy implemented in 2013) on DAPP[58]. In the future, strategies to meet SDG3.9 can be explored by considering specific policy implementations[29,38] and using detailed emissions inventories[17,35,59].

Finally, we set a series of standards for all countries (i.e., 10, 20, and 30%) to reflect the challenges of meeting SDG3.9 globally. Although they are uniform in proportion, the absolute numbers of DAPP are country-specific. Similar to the air quality targets proposed by WHO (2021)[60], countries can further specify their own targets of

SDG3.9 based on our analyses to suit their socioeconomic development stage. When analyzing the additional improvement in healthcare and air pollution control, we used simple scenarios which generalize the complex interactions among socioeconomic development, healthcare, and aging. Policy efforts towards lowering $PM_{2.5}$ concentrations and improving baseline mortality rates are far more achievable via policymaking than other measures for reducing DAPP such as lowering population aging and hence, these intervention scenarios can provide general insights into the scale of response of DAPP required to meet SDG3.9. This simplification should be interpreted as a sensitivity analysis and may underestimate DAPP. More specific target setting and exploration of pathways by country considering emissions, energy structure, socioeconomic development, and political factors are needed[38,61].

## Methods

### Scenario framework

Part of the Coupled Model Intercomparison Project Phase 6 (CMIP6), ScenarioMIP is a new, widely accepted scenario framework[62,63] which provides integrated, internally consistent ensemble simulations of driving factors suitable for research across multiple scientific fields (Table 1). ScenarioMIP represents a matrix of possible integration of multiple SSPs and forcing outcomes driven by a set of emissions and land use scenarios, produced with integrated assessment models (IAMs)[64]. Compared with previous RCPs derived from earlier emissions and land use scenarios[65], ScenarioMIP scenarios provided a related but updated simulation.

In ScenarioMIP, eight representative scenarios were provided to describe possible futures, which can be further divided into Tier 1 and Tier 2 based on relative priority[62]. Considering the accessibility of data, we used the four Tier 1 scenarios: Sustainability (SSP1-2.6), representing sustainable development in both socioeconomic and environmental aspects; Middle of the Road (SSP2-4.5), representing the continuation of recent global trends; Regional Rivalry (SSP3-7.0), representing a world of high inequality in human and economic opportunities, and Fossil-fueled Development (SSP5-8.5), representing a future with prosperous socioeconomic development embodied with improved air quality control at the expense of climate, with high energy demand[66].

Under this framework, studies have projected the corresponding socioeconomic factors and concentration of pollutants[50,67,68], with stricter air pollution controls tied to higher levels of economic development[66]. Hence, weak air pollution controls occur in SSP3-7.0, with medium controls in SSP2-4.5, and strong air pollution controls in SSP1-2.6 and SSP5-8.5[25]. These studies provided a quantitative and systematic foundation for the robust and internally consistent projection of future trajectories of global DAPP. The corresponding data under ScenarioMIP includes surface concentration of pollutants ($SO_4$, black carbon, organic aerosol, dust, sea salt), age-specific population, and socioeconomic data (fertility, GDP per capita, and average years of education)[67,68] (Supplementary Note 1).

### Estimating deaths attributable to $PM_{2.5}$ pollution

Historical (2000–2015) DAPP can be estimated based on total population, population age structure, measured annual average $PM_{2.5}$ concentration, as well as the death rate of diseases (details in Supplementary Note 1). The equation is given as follows[4,5]:

$$DAPP = \sum_{a,d} (PAF_{a,d} \times POP \times Rate_{a,d} \times AgeP_a) \quad (1)$$

where $DAPP$ is the deaths attributable to $PM_{2.5}$ pollution; $PAF_{a,d}$ is the proportion of deaths attributed to $PM_{2.5}$ pollution caused by disease $d$ in a population with age $a$; $POP$ refers to the total population; $Rate_{a,d}$ is the death rate of disease $d$ for people with age $a$, and $AgeP_a$ is the

percentage of the total population of age $a$. In our formulation, 15 age groups were included in the equation, i.e., 25–30, 30–35, ..., 90–95, and 95+ years old. Six diseases related to $PM_{2.5}$ pollution were considered in this study, including lung cancer, chronic obstructive pulmonary disease, lower respiratory infection, ischemic heart disease, stroke, and diabetes mellitus type 2. Evidence linking these diseases with exposure to ambient air pollution was judged to be consistent with a causal relationship on the basis of criteria specified for Global Burden of Disease (GBD) risk factors, including meta-analysis, cohort study, and biologically plausible relationship[5]. Although $PM_{2.5}$ pollution is also related to other adverse birth outcomes including low birth weight and short gestation, these 6 diseases represent around 95% of total deaths related to $PM_{2.5}$ pollution from all causes based on the estimation of GBD 2019[5]. We did not consider differences by gender to reduce the complexity of our projections.

$PAF_{a,d}$ refers to the proposition that a given disease proportion of deaths attributed to $PM_{2.5}$ pollution caused by disease $d$ in a population with age $a$, which can be calculated as below[69].

$$PAF_{a,d} = \frac{RR_{a,d} - 1}{RR_{a,d}} \quad (2)$$

where $RR_{a,d}$ is the relative risk for the population with age $a$ of acquiring disease $d$. This refers to the ratio of incidence for an exposed population compared to an unexposed population. Relative risk can be quantified based on a non-linear exposure-response function which usually shows a steeper increase at lower concentrations with more modest increases at higher concentrations. In this study, we used the latest meta-regression (i.e., MR-BRT) exposure-response functions updated by GBD 2019[5] (Figure S3). For ischemic heart disease and stroke, we used age-specific exposure-response functions because the epidemiological evidence suggests that the relative risks for these diseases change by age[4,49]. For the other four diseases, the exposure-response functions are uniform for all age groups. Details of the calculation and validation of DAPP can be found in Supplementary Note 2.

### Projecting future deaths attributable to $PM_{2.5}$ pollution

We projected DAPP under different scenarios from 2015 to 2050 by introducing the corresponding driving factors from the ScenarioMIP framework into the above-mentioned epidemiological model. Among the driving factors used for the projection of DAPP, the total population and age structure were derived from existing projections in the SSPs database, consistent with the ScenarioMIP framework[68] Future $PM_{2.5}$ concentration and the death rate of diseases were both estimated in this study.

We used data from 11 climate and earth system models in the CMIP6 database to estimate future $PM_{2.5}$ concentrations. Eight models are available for all four scenarios. In addition, one model (GFDL-CM4) provided data for SSP2-4.5 and 5-8.5 only, and two models (BCC-ESM1 and CNRM-ESM2-1) were only available for SSP3-7.0 (Table S3). Under a given scenario, these models use consistent anthropogenic and biomass-burning emissions from the same dataset[25,66], but differ in other natural emissions (e.g., dust, biogenic volatile organic compounds, and others.) and aerosol scheme[50]. For example, only GISS-E2-1-G and GFDL-ESM4 provide ammonium and nitrate mass mixing ratios. For the CNRM-ESM2-1 model, anomalously large concentrations were obtained from sea salt mass mixing ratios. To ensure consistency, we calculated $PM_{2.5}$ concentration offline with surface concentration of pollutants via the below equation:

$$PM_{2.5} = BC + OA + SO_4 + NH_4 + 0.25 \times SS + 0.1 \times dust \quad (3)$$

where $BC$ is black carbon; $OA$ is organic aerosols (primary and secondary) and $SS$ is sea salt. The concentration of $NH_4$ is not estimated in

all CMIP6 models so we estimated it as $NH_4 = (36 \times SO_4)/96$ assuming that $NH_4$ is only present as ammonium sulfate[20]. The factors 0.25 and 0.1 were intended to approximate the fractions of sea salt and dust in the PM$_{2.5}$ size range. As one component of PM$_{2.5}$, nitrate was only reported by two models from CMIP6, therefore, following Silva et al.[70] and Silva et al.[71], we omitted it from the PM$_{2.5}$ concentration formula to avoid inconsistencies with other CMIP6 models. Where models included multiple ensembles, a mean was taken using all available members for each model.

Because the PM$_{2.5}$ concentration from the CMIP6 models and historical estimation was not comparable, we further calibrated PM$_{2.5}$ concentration based on the trend in simulated PM$_{2.5}$ concentration relative to historical PM$_{2.5}$ concentration measurements for the base year 2015. Please see Supplementary Note 3 and Figs. S5–6 for the details of PM$_{2.5}$ concentration projection, validation, and uncertainty analysis.

We estimated future death rate of diseases based on the historical death rate of diseases and future socioeconomic development indicators using the model developed by Foreman et al.[11]. This model assumed the cause-specific death rate would change with socioeconomic development following the formulae:

$$ln(m) \sim N(\hat{y} + \hat{\epsilon}, \sigma) \tag{4}$$

$$\hat{y} = \beta_1 SDI_{<0.8} + \beta_2 SDI_{\geq 0.8} + \theta_a t + \alpha_{la} + ln(R) \tag{5}$$

$$\hat{\epsilon} = ARIMA\left(\epsilon_{history}\right) \tag{6}$$

where $ln(m)$ refers to the natural logarithm of the cause-specific death rate, which can be determined by $\hat{y}$ (representing the effects of the long-term trend, socioeconomic development, and risk factors) and $\hat{\epsilon}$ (representing the residuals that cannot be explained by these factors). In the equation to calculate $\hat{y}$, $\beta_1$ and $\beta_2$ represent the global coefficients of the Socio-Demographic Index (SDI) when its value is less or more than 0.8. $\theta_a$ refers to an age-specific secular trend, and $t$ represents time. $\alpha_{la}$ is the region and age-specific intercept, and $ln(R)$ represents a scalar that captures the effects of risk factors. SDI is calculated by combining the logarithm of income per person, educational attainment, and total fertility rate under 25 years. $\hat{\epsilon}$ can be derived using historical data based on the autoregressive integrated moving average model (ARIMA). The detailed projection, validation, and uncertainty analysis for the future death rates of diseases can be found in Supplementary Note 4 and Figures S7-8. The global and regional long-term trends of underlying drivers used to project DAPP are presented in Figures S9-10.

We then quantified the uncertainty in the projection of DAPP. The driving factors and the epidemiological model both introduce uncertainty into DAPP projection. We considered the range of uncertainty in future PM$_{2.5}$ concentration and the death rate of diseases, and calculated the possible range in DAPP based on these uncertainty intervals. Regarding the epidemiological model, uncertainty mainly comes from the choice of the exposure-response function. We present the results derived from the medium MR-BRT function, and conducted a similar uncertainty analysis based on the upper (97.5[th]) and lower (2.5[th]) estimates of MR-BRT to validate the robustness. Detail of the uncertainty in DAPP projections can be found in Supplementary Note 2.

### Measuring the attainment of SDG3.9

SDG3.9 commits countries to a substantial reduction in the number of deaths and illnesses from hazardous chemicals and air, water, and soil pollution and contamination[7]. However, the term substantial reduction is subjective and not clearly defined in SDG3.9. Previously, GBD2017 SDG Collaborators (2018)[72] used the top 10[th] percentile of

performance among country-level rates of change before 2015 as the annualized change rate of indicators between 2015 and 2030 required by 25 SDG targets. Hence, targets which are not quantitatively defined can be specified by assuming their progress is keeping pace with defined indicators. We set quantitative targets for SDG3.9 based on the 10th percentile reduction in DAPP at the national scale during 2000–2015, and the estimated value is around 30%. In addition, to acknowledge the different levels of target setting and the uncertainty of pathways, following Moallemi et al.[27], we roughly defined the substantial reduction in terms of three levels of ambition in DAPP reduction targets under SDG3.9, where weak, moderate, and ambitious targets are represented by a 10%, 20%, and 30% reduction compared to 2015 levels, respectively.

### Quantifying the effect of different driving factors on the attainment of SDG3.9

DAPP is a function of the nonlinear interaction (Eq. 1) of different driving factors (population size, PM$_{2.5}$ concentration, age structure, and death rate of diseases). Hence, a decomposition method[4,73] was taken to dissect the contributions of these factors to the change in DAPP. The decomposition method estimates the contribution of factors by sequentially introducing each factor into the DAPP equation. The difference between each consecutive step represents the relative contribution of the corresponding factor. As the sequence of adding factors also influences the results, we estimated the results under all 24 possible sequences of the four factors. The final estimation of contributions from different factors is the average value of the results for each factor. Detailed equations and processes are shown in Figure S18.

### Exploring pathways to meet SDG3.9 via additional improvements in air pollution control and healthcare

Based on ScenarioMIP projection, we also considered the potential of additional efforts to reduce DAPP. Among the four components influencing DAPP, demographic factors (population and age structure) cannot be effectively altered by policy intervention in the short term, but PM$_{2.5}$ concentration and death rate of diseases can more readily be changed via additional efforts in air pollution control (e.g., stricter air quality standards, technological innovation, and cleaner energy mix)[38,61] and healthcare (e.g., investment in medical research, scientific breakthroughs, health system improvements, and better accessibility)[11,74]. We considered three possible conditions, including additional air quality improvement (20% lower PM$_{2.5}$ concentration + No change in death rate of diseases); additional healthcare improvement (No change in PM$_{2.5}$ concentration + 20% lower death rate of diseases), and both measures (20% lower PM$_{2.5}$ concentration + 20% lower death rate of diseases). The value of 20% was selected subjectively to represent a substantial and plausible improvement resulting from concerted government attention. With these assumptions, we conducted a sensitivity analysis to assess the response of additional air quality and healthcare interventions towards SDG3.9.

### Reporting summary

Further information on research design is available in the Nature Portfolio Reporting Summary linked to this article.

## Data availability

All the data generated in this study have been deposited in the GitHub repository https://github.com/yuehuanbi/attainment-of-SDG3.9. Other specific data are available from the corresponding author upon reasonable request.

## Code availability

We used the integrated exposure-response function updated in the Global Burden of Disease 2019 to estimate the relative risk caused by PM$_{2.5}$ exposure. The detailed function is accessible to all users at

https://ghdx.healthdata.org/record/global-burden-disease-study-2019-gbd-2019-air-pollution-exposure-estimates-1990-2019. Other data processing in this study are conducted using ArcGIS and Microsoft Excel, see details in Supplementary Note 2-4.

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

## Acknowledgements

We express our gratitude to the anonymous reviewers and editors for their professional comments and suggestions. We thank Prof. Samir K.C. (International Institute for Applied Systems Analysis, Laxenburg, Austria) and Dr. Steven T. Turnock (Met Office, Exeter, UK) for their generous sharing of data. We would like to thank the high-performance computing support from the Center for Geodata and Analysis, Faculty of Geographical Science, Beijing Normal University [http://gda.bnu.edu.cn/]. C.H. was funded by the National Natural Science Foundation of China (Grant No. 42371296) and BNU-FGS Global Environmental Change Program (No. 2023-GC-ZYTS-08). H.Y. was supported by the Shandong Provincial Natural Science Foundation (ZR2022QD051). D.Z. received funding from the National Natural Science Foundation of China (Grant No. 42271314). Q.H. was supported by the and Beijing Nova Program (Grant No. 20220484163) and B.B. was supported by Deakin University, Australia.

## Author contributions

H.Y., C.H., Q.H. and B.B. designed the study and developed the analysis plan. H.Y., B.B., Y.Y., X.Q. and D.Z. prepared the basic data and did the data analysis and visualization. H.Y., Q.H., B.B., Q.M., E.M. and D.Z. drafted the manuscript. Q.H., F.X. and P.S. verified the underlying data. All authors contributed to the interpretation of findings, provided revisions to the manuscript, and approved the final manuscript.

## Competing interests

We declare no competing interests.
