## [Peer Review File · Nature Communications]

Substantially reducing global PM2.5-related deaths under SDG3.9 requires better air pollution control and healthcareREVIEWER COMMENTS

Reviewer #1 (Remarks to the Author):

This paper presents future trends in premature death attributable to ambient PM_{2.5} exposure under four representative SSP scenarios considering both changing PM_{2.5} levels from 11 models in CMIP6 and socio-demographic factors. Although the topic is of interest, much more effort is needed in data analysis and results interoperation to clarify this study's new findings and major contributions. I think the authors need to clarify a number of major issues if they want this paper to be well received. I also noted multiple errors in the text and figures, which shows the manuscript's quality needs improvement.

General comments

1. This study's major contribution would be the projection of DAPP using all 11 available simulations of PM_{2.5} from CMIP6. But only projections based on the average values and 95% CI of all simulations are analyzed. The future trends of PM_{2.5} between CMIP6 models differ in both the magnitudes and signs because of their differences in natural emissions, chemical mechanisms, etc. The substantial impacts of variations in PM_{2.5} trends can also be seen in Figure 6. The large uncertainty ranges mean whether the SDG3.9 target can be attained or even the sign of future changes in DAPP largely depends on the choice of models. What models are included in each scenario? How does the difference in model prediction of PM_{2.5} trends impact DAPP trends and the attainment of SDG3.9 target? What factors are driving the differences? Is it emissions (natural emissions, I guess) or representation of the chemical/physical processes in the models?
2. A related comment would be that the authors need to prove that the ensemble average of all models is superior to a specific model in CMIP6 in predicting PM_{2.5} trends and DAPP. To me, some models are oversimplified in their aerosol schemes. An approximation is needed to derive PM_{2.5} concentrations from all 11 models (equation 3). Is such an approximation better than the results from one or two model simulations with more detailed aerosol schemes? Ammonium nitrate may contribute more to the PM_{2.5} mass than ammonium sulphate in the future, especially in scenarios where coal-fired power plants are aggressively phased out.
3. The section "effects of individual factors on the attainment of SDG3.9" is another highlight of this study. Regrettably, the discussion on the driving factors is oversimplified. Region- and country-specific decomposition results in the SI figures without any discussion.

Specific comments

1. Figure 1: the schematic figure for PM_{2.5} exposure looks like one for infection from respiratory viruses through mouth.
2. Line 162: Do you mean colored dots and hollow dots? This figure is hard to read. The information contained in this figure is better summarized in an SI table.
3. Recheck the text. I noted multiple unwanted words (e.g., lines 91 and 121).

Reviewer #2 (Remarks to the Author):

This study conducts a comprehensive analysis of future projected impacts of PM_{2.5} on health outcomes under different climate change scenarios. The study uses state-of-the-art methods, is well written and can be extremely useful for policy implications. There are a number of comment below that would need to be addressed.

Major comments:

Can authors elaborate on why they used these three specific levels of ambitions: (i.e. 10, 20, and 30% reduction compared to 2015 levels)?

Authors discuss adequately the limitations of this study. They should conduct sensitivity analyses

when considering future projections in joint exposures to temperature and PM2.5 or they should at least discuss it.

Please provide further rationale on why those 6 specific diseases related to air pollution were evaluated.

Exposure-response functions were used and derived from published meta-regression models. This is appropriate. However, exposure-response functions do not appear to account for age-group specific associations as well as gender specific associations. Also, what about providing projections by age group and gender? Can authors clarify?

Projections are conducted across large geographical areas. What about the importance of evaluating sub-national projections?

This study specifically focused on PM2.5 projections, but there could also be other pollutants affected such as ozone. Can authors comment on this issue?

Projections that were provided in this study looked specifically at long term effects of PM2.5. There are also concerns regarding its short-term effects. Can this be incorporated into the analysis to obtain a better overall impact of the issue. Otherwise, please comment on this issue.

Impacts of PM2.5 on health outcomes can be non-linear (or even supralinear) where the curve usually shows a steeper increase at lower concentrations and flattens out at higher concentrations. Please comment on how this would affect the future projections of PM2.5 on health.

Minor comments:

Line 91 : I believe it should read "and the results are limited in capturing the range..."

Line 121: This part if repeated twice: "a great challenge".

Reviewer #3 (Remarks to the Author):

GENERAL COMMENTS:

The study analyses the potential reduction of premature mortalities attributable to ambient PM2.5 (DAPP) under alternative SSP-RCP scenarios, and if/how do they align with the "substantial" reduction mentioned by the UN's SDG 3.9. The manuscript studies the influences of four key drivers (PM2.5 concentration levels, baseline mortality rates of diseases, population growth, and aging) for the evolution of future DAPPs for different regions over the world. The topic is of interest for the scientific community and for non-expert stakeholders, as it shows the need for more ambitious action to obtain a significant reduction in premature mortalities associated with air pollution.

I have some general comments. First, I think the contribution of the study could be reframed. There exist several global and regional studies that assess the reduction of DAPPs for different scenarios which already consider dynamic PM2.5 concentration levels, population structure and aging, or baseline mortality rates. Moreover, these studies include a wide range of scenarios that go beyond the standard SSP-RCP scenarios from ScenarioMIP. Likewise, the study also explores the effects of additionally reducing PM2.5 levels and improving the healthcare system by 20%, an arbitrary magnitude that drives to an expected or obvious result. On the other hand, the study provides some interesting results, such as the gridded or city-level analysis of the future reduction in DAPPs, or the decomposition analysis. I believe these should be the main focus of the paper in order to make it more relevant for the community.

I also have some concerns related to some of the assumptions that should be further explained. First,

the study assumes that investing in healthcare would directly reduce the baseline mortality rates, which directly decreases DAPPs. This is important since it seems to be the strongest factor for decline of DAPPs (Line 178). This causality is not very obvious and I think it should be further elaborated considering its implications. Furthermore, Table 1 shows that the emission controls in the SSP5-8.5 scenario (development of FF) are similar to the controls in the Sustainability scenario (SSP1-2.6). Is this an assumption on ScenarioMIP? If not, this can be counterintuitive (and incorrect) and it should be explained in more detail.

Finally, regarding the presentation of the results, I believe that description of some of the results is too vague (particularly in the first results subsection) and too descriptive without getting deeper into the reasons or drivers. In addition, the discussion is quite repetitive. My recommendation for the discussion would be to take the first paragraphs out, and mix the points raised from line 288 with the limitations (line 325)

Overall, the study comprises some interesting information that would be relevant for the community, but I believe it should be substantially reframed, in order to focus on those aspects (see above) that would be the most relevant for the scientific community and expert and non-expert stakeholders.

SPECIFIC COMMENTS:

- (Line 78) I think there is a substantial body of literature exploring AAP-related health impacts, also for SSP and decarbonization scenarios. The comparison should include more recent studies that have not been considered.
- (Line 99) The study explores SSP-RCP combinations, not just SSP narratives.
- (Line 121) Duplicated "a great challenge".
- (Line 157) I think the information in Figure 3 is very powerful, but the format of the figure is confusing... I would try to find another way of presenting these (interesting) results.
- (Line 201) Again, I am not sure that investment in healthcare reduces the disease rates, at least in some regions.
- (Line 227). As indicated, it is strange that, with no further justification, the Fossil-fuel-dependent scenario reduces more the PM2.5 concentration levels than other scenarios due to stringent air pollution controls...
- (Line 313). It is quite obvious that more stringent controls and improving healthcare will reduce PM2.5-related mortalities... I think this discussion is vague, and should consider other aspects such as costs, technological feasibility etc.
- (Line 357): It is not clear the distinction between SSP narratives and RCPs.
- (Line 463): Not additional explanation why 20% has been chosen.

Detailed author response to reviewer comments: NCOMMS-23-06278

Yue et al. "Substantially reducing global deaths from PM_{2.5} pollution under SDG3.9 requires advances in both air pollution control and healthcare"

We would like to express our gratitude to the editor and anonymous reviewers for their valuable comments and suggestions for improving the quality of the paper. We have carefully considered all the points raised by them. We are providing detailed point-by-point responses to all questions and recommendations by the reviewers. In the responses below, red fonts are the revised texts.

Response to the reviewer 1

This paper presents future trends in premature death attributable to ambient PM_{2.5} exposure under four representative SSP scenarios considering both changing PM_{2.5} levels from 11 models in CMIP6 and socio-demographic factors. Although the topic is of interest, much more effort is needed in data analysis and results interoperation to clarify this study's new findings and major contributions. I think the authors need to clarify a number of major issues if they want this paper to be well received. I also noted multiple errors in the text and figures, which shows the manuscript's quality needs improvement.

Issue #1-1: This study's major contribution would be the projection of DAPP using all 11 available simulations of PM_{2.5} from CMIP6. But only projections based on the average values and 95% CI of all simulations are analyzed. The future trends of PM_{2.5} between CMIP6 models differ in both the magnitudes and signs because of their differences in natural emissions, chemical mechanisms, etc. The substantial impacts of variations in PM_{2.5} trends can also be seen in Figure 6. The large uncertainty ranges mean whether the SDG3.9 target can be attained or even the sign of future changes in DAPP largely depends on the choice of models. What models are included in each scenario? How does the difference in model prediction of PM_{2.5} trends impact DAPP trends and the attainment of SDG3.9 target? What factors are driving the differences? Is it emissions (natural emissions, I guess) or representation of the chemical/physical processes in the models?

Response #1-1: Revised as suggested.

Following the reviewer's suggestion, we extended the details in the **Methods** section. First, we specified how many models and their corresponding scenarios were included in this study in a supplementary table (Table S1).

The revised version was shown in line 477-485 as follows:

“We used data from 11 climate and earth system models in the CMIP6 database to estimate future PM_{2.5} concentrations. Eight models are available for all four scenarios. In addition, one model (GFDL-CM4) provided data for SSP2-4.5 and 5-8.5 only, and two models (BCC-ESM1 and CNRM-ESM2-1) were only available for SSP3-7.0 (Table S3). Under a given scenario, these models use consistent anthropogenic and biomass-burning emissions from the same dataset^{1,2}, but differ in other natural emissions (e.g., dust, biogenic volatile organic compounds, and others.) and aerosol scheme³. For example, only GISS-E2-1-G and GFDL-ESM4 provide ammonium and nitrate mass mixing ratios. For the CNRM-ESM2-1 model, anomalously large concentrations were obtained from sea salt mass mixing ratios.”

Second, to investigate how the difference in model prediction of PM_{2.5} trends impact DAPP trends and the attainment of SDG3.9 target, we added texts and figures besides the average results. In specific, we also compared the results derived from multiple CMIP6 models, and added new elements (Figure S4, Figure S9, Table 2) to illustrate the difference in PM_{2.5} concentrations, DAPP and attainment of SDG3.9 target in the **Trends in DAPP under different scenarios and the attainment of SDG3.9** section in the **Results** section. The revised version was shown in line 165-175 as follows:

“Future trends in PM_{2.5} concentrations among different CMIP6 general circulation models differed in both magnitude and sign because of their differences in natural emissions, chemical mechanisms, and processes. The multi-model average DAPP was estimated based on all available models to provide a general trend (see details in Methods). To further encompass the uncertainties among the 11 models, we also calculated the model-specific trend in DAPP (Figure S9) and found that most models yielded results similar to the average estimates (calculated based on all available models). Although some models (e.g., MIROC-ES2L, INM-CM5-0) displayed a steeper decline in DAPP, the moderate SDG3.9 target was not achieved by 2030 for any scenario except SSP1-2.6 (Figure S4; Table 2). Considering some models were scenario-specific, to ensure consistency, we also presented the average results calculated based on the eight models with estimates for all scenarios, which was highly in line with average results from all available models.”

Fig. S1 Trends of deaths attributable to PM_{2.5} pollution by model. The results were estimated based on medium value of exposure-response function and death rate of disease. Please refers to Fig. S1 for detailed definitions of ASIA, OECD, MAF, REF and LAM. Please refers to Table S3 for detailed information of models.

Fig. S2 Trends of population-weighted $PM_{2.5}$ concentration by model. The $PM_{2.5}$ concentration is weighted by population to indicate the overall exposure. Please refers to Fig. S1 for detailed definitions of ASIA, OECD, MAF, REF and LAM. Please refers to Table S3 for detailed information of models.

Table 2. Changes in deaths attributable to PM_{2.5} pollution (DAPP) under different CMIP6 general circulation (climate) models. The DAPP was estimated based on ensemble average PM_{2.5} concentration from different models. Light blue, blue and dark blue indicates weak, moderate and ambitious targets of SDG3.9 respectively.

Change in DAPP relative to 2015 (%)	Scenarios	Models												
		Average	Average-8 models	GFDL-ESM4	GISS-E2-1-G	INM-CM4-8	INM-CM5-0	MIROC-ES2L	MRI-ESM2-0	NorESM2-LM	NorESM2-MM	GFDL-CM4	BCC-ESM1	CNRM-ESM2-1
By 2030	SSP1-2.6	-18.9	-18.7	-20.0	-16.4	-19.9	-20.8	-22.5	-18.3	-16.0	-17.1	N/A	N/A	N/A
	SSP2-4.5	-0.9	-0.7	-2.5	-0.6	-1.4	-2.8	-2.5	0.1	0.9	1.4	-0.4	N/A	N/A
	SSP3-7.0	13.8	13.4	15.2	16.1	10.8	11.1	14.7	13.7	14.8	12.6	N/A	14.3	14.2
	SSP5-8.5	-11.0	-10.8	-11.0	-12.6	-13.1	-13.9	-14.0	-10.5	-5.4	-6.2	-12.6	N/A	N/A
By 2050	SSP1-2.6	-24.6	-24.9	-22.8	-19.1	-36.3	-38.7	-28.4	-20.8	-15.9	-15.0	N/A	N/A	N/A
	SSP2-4.5	1.2	1.1	0.4	3.6	-6.1	-6.4	-2.6	7.1	6.3	6.0	2.7	N/A	N/A
	SSP3-7.0	31.4	30.6	35.6	32.5	28.3	26.9	30.8	32.4	32.6	30.7	N/A	31.8	32.0
	SSP5-8.5	-7.3	-7.2	-5.3	-5.6	-15.5	-16.2	-12.5	-4.4	4.9	-0.9	-10.4	N/A	N/A

Attainment of SDG3.9 for nations by different CMIP6 models were also presented in line 191-202 as follows:

Figure 1. Attainment of SDG3.9 by 2030 and 2050 for the 154 nations. The colors indicate percentages of countries that can meet SDG3.9 under 0, 1, 3, 2, and 4 possible scenarios assessed with, weak, moderate and ambitious settings for SDG3.9 represented as 10%, 20% and 30% reduction in DAPP relative to 2015, respectively. Only 8 models that were available for all the 4 scenarios were included in the model-specific analysis.

“While considering the variations among different models, the challenge of meeting SDG3.9 was still robust that considerable amount countries cannot achieve SDG3.9 under any scenarios. Even we assessed the results with the relatively loose standard (10% reduction) based on the most ideal model in PM_{2.5} pollution reduction (MIROC-ES2L), more than one-third of nations failed to meet SDG3.9. The ratio became 55% (moderate) and 69% (ambitious) when tightening the target.”

At last, we further discussed the reason of difference of PM_{2.5} concentration from multiple CMIP6 models in the **Discussion** section. The revised version was shown in line 383-390 as follows:

“Regarding PM_{2.5} concentration, there are several potential explanations for the discrepancy between the 11 CMIP6 models, including differences in the treatment of aerosols and their components (e.g., organic aerosols and emission of biogenic volatile organic compounds) as well as the effect of climate change (i.e., temperature and precipitation) simulated by models and its impact on natural aerosol emissions³. To account for this variation, we presented model-specific (Figure S9) as well as multi-model average results with ranges (Figure S15-16) reflecting uncertainty in future PM_{2.5} concentration.”

Issue #1-2: A related comment would be that the authors need to prove that the ensemble average of all models is superior to a specific model in CMIP6 in predicting PM_{2.5} trends and DAPP. To me, some models are oversimplified in their aerosol schemes. An approximation is needed to derive PM_{2.5} concentrations from all 11 models (equation 3). Is such an approximation better than the results from one or two model simulations with more detailed aerosol schemes?

Response #1-2: Clarified and Revised.

The choice of model did influence our results as the reviewer suggested. However, it's beyond our scope to say which model is superior in the simulation of PM_{2.5} concentrations. Hence, we still used the multi-model average concentrations to depict the general trends of PM_{2.5} concentration and DAPP. Considering the variations resulting from the 11 models, we also presented model-specific results besides the multi-model average results, and discussed the uncertainty (as our response#1-1). We added more information in the **Methods** section to acknowledge the differences in CMIP6 models' aerosol schemes and other information. The revised version was shown in line 479-485 as follows:

“Under a given scenario, these models use consistent anthropogenic and biomass-burning emissions from the same dataset^{1,2}, but differ in other natural emissions (e.g., dust, biogenic

volatile organic compounds, and others.) and aerosol scheme³. For example, only GISS-E2-1-G and GFDL-ESM4 provide ammonium and nitrate mass mixing ratios. For the CNRM-ESM2-1 model, anomalously large concentrations were obtained from sea salt mass mixing ratios. To ensure consistency, we calculated PM_{2.5} concentration offline with surface concentration of pollutants via the below equation...

Issue #1-3: The section "effects of individual factors on the attainment of SDG3.9" is another highlight of this study. Regretfully, the discussion on the driving factors is oversimplified. Region- and country-specific decomposition results in the SI figures without any discussion.

Response #1-3: Revised as suggested.

We reorganized the **Results** section on decomposition to emphasize its importance. First, we explained the effects of individual factors on attainment of SDG3.9 at the global scale and country-specific scale using the following three paragraphs in line 216-251. We also used Figure S19-20 to present region- and country-specific decomposition results.

“Population aging was the most significant factor under all scenarios in driving growth in DAPP, causing an increase of 34% (SSP3-7.0) to 41% (SSP5-8.5) by 2030. Conversely, healthcare improvement was the strongest factor driving down DAPP, leading to a decline of 36% (SSP3-7.0) to 44% (SSP5-8.5). As socioeconomic development reduced the death rate of diseases and exacerbated population aging simultaneously, these two drivers tended to offset each other, resulting in a relatively minor net decrease of around 1 to 3% (Figure 5). This counteracting effect also influenced the age distribution of DAPP. The share of DAPP amongst older people (65+) increased from almost 65% in 2015 to around 70% under all scenarios by 2030 due to the increase in older adults and higher life expectancy occurred alongside better healthcare (Figure S5). Population growth led to a minor increase in DAPP of around 6% (SSP1-2.6) to 14% (SSP3-7.0).

Compared with other driving factors, the effect of PM_{2.5} concentration was more variable across scenarios. For instance, DAPP in 2030 was projected to decrease by 24% relative to 2015 because of the decline in PM_{2.5} concentration alone under SSP1-2.6, led by the effort in climate mitigation and strong air pollution control. Similarly, air quality improvement was responsible for a decline of 15% and 8% in DAPP under SSP5-8.5 (strong air pollution control and high energy demand) and SSP2-4.5 (moderate air pollution control and moderate energy demand), respectively. Under the worst scenario for air pollution (SSP3-7.0) which assumes a weak air pollution control with slow socioeconomic development, DAPP was projected to increase by 2% due to a 12% increase in PM_{2.5} concentration relative to 2015 (Figure 5; Figure S15). As the relationship between PM_{2.5} concentration and health effects represented by exposure-

response curves is non-linear, DAPP tended to show a steeper increase at lower concentrations and levelled off at higher concentrations (Figure S3). Hence, a lower level of PM_{2.5} concentration in the future would substantially reduce DAPP. For example, under SSP1-2.6 a decrease in PM_{2.5} concentration of 31% led to a 24% decrease in DAPP from 2015-2030; whereas under SSP5-8.5, a growth of 12% in PM_{2.5} concentration led to a 2% increase in DAPP (Figure 5).

Similar patterns in driving factors occurred beyond 2030 at the global scale. Aging and healthcare improvement remained the dominant driving factors affecting DAPP and the attainment of SDG3.9 while the effect of PM_{2.5} concentration varied among scenarios. The effect of population growth between 2030-2050 was smaller relative to that from 2015-2030 because population growth was slower. Detailed region- and country-specific decomposition were also conducted (Figure S19-20). For most countries, changes in aging and healthcare were the main contributors to DAPP.”

In addition, we also extended the discussions of different driving factors in the **Discussion** section to acknowledge the interactions among factors. The revised version was shown in line 309-333 as follows:

“Global DAPP will not be substantially reduced under all but the most ambitious scenarios (SSP1-2.6) because of the overwhelming effect of population aging (Figure 1). However, there are complex interactions between different scenarios. For instance, with socioeconomic development, the population will get older which increases the size of vulnerable populations, but the death rate of diseases declines which simultaneously reduces population vulnerability. Hence, the effect of population aging will be offset by improvement in healthcare, hence scenarios with high population aging (SSP1-2.6 and SSP5-8.5) showed greater reduction in DAPP. Besides the improvement in healthcare, changes in PM_{2.5} concentration are also a key driver of reducing DAPP. SSP1-2.6 had the largest reduction because of the underlying assumption of a cleaner energy mix and strong air pollution control. However, PM_{2.5} concentration is also expected to reduce in SSP5-8.5. That is because of the strong air pollution control led by the adoption of the current best available technology (especially end-of-pipe control) and socio-economic development based on high energy demand and fossil fuel dominated energy mix. Hence, transformation aligned with an SSP1-2.6 future with stronger air quality control and less fossil fuel dependence is the better pathway for reducing DAPP (Figure 5, Figure S9) as it has positive co-benefits for many other SDGs that are not assessed in this study.

In addition, the interaction between different drivers varied by region. For instance, besides population aging, population growth in MAF also played a considerable role under all

scenarios because the region’s population is relatively young and growing fast. While in LAM and ASIA, aging is the dominant driver of growth in DAPP (Figure S19). Different countries also exhibited heterogeneity in mechanisms driving DAPP. For instance, China is likely to achieve SDG3.9 under most scenarios because of the reversal of population growth and the declining trend of PM_{2.5} pollution, as well as healthcare improvement. While for India, DAPP tended to increase under three scenarios (all bar SSP1-2.6) mainly because of the joint effect of air quality deterioration and demographic change.”

Issue #1-4: Figure 1: the schematic figure for PM_{2.5} exposure looks like one for infection from respiratory viruses through mouth.

Response #1-4: Revised as suggested.

We have modified the schematic figure to emphasize the infection from respiratory by mouth and nose. The shape of PM_{2.5} was also modified. The revised version was shown in line 88-90 as follows:

Figure 2. The impact of socioeconomic development and climate change on deaths attributable to PM_{2.5} pollution.

Issue #1-5: Do you mean colored dots and hollow dots? This figure is hard to read. The information contained in this figure is better summarized in an SI table.

Response #1-5: Revised as suggested.

We redesigned figure 3 in a more intuitive way. The revised version was shown in line 191-196 as follows:

Figure 3. Attainment of SDG3.9 by 2030 and 2050 for the 154 nations. The colors indicate percentages of countries that can meet SDG3.9 under 0, 1, 3, 2, and 4 possible scenarios assessed with, weak, moderate and ambitious settings for SDG3.9 represented as 10%, 20% and 30% reduction in DAPP relative to 2015, respectively. Only 8 models that were available for all the 4 scenarios were included in the model-specific analysis.

Issue #1-6: Recheck the text. I noted multiple unwanted words (e.g., lines 91 and 121).

Response #1-6: Revised as suggested.

We rechecked the text. The entire manuscript has been edited for brevity and clarity.

Response to the reviewer 2

This study conducts a comprehensive analysis of future projected impacts of PM_{2.5} on health outcomes under different climate change scenarios. The study uses state-of-the-art methods, is well written and can be extremely useful for policy implications. There are a number of comments below that would need to be addressed.

Issue #2-1: Can authors elaborate on why they used these three specific levels of ambitions: (i.e., 10, 20, and 30% reduction compared to 2015 levels)?

Response #2-1: Revised as suggested.

We clarified the reasons why we used these three specific levels of ambitions in the **Methods** section. The revised version was shown in line 528-538 as follows:

“However, the term *substantial reduction* is subjective and not clearly defined in SDG3.9. Previously, GBD2017 SDG Collaborators (2018)⁴ used the top 10th percentile of performance among country-level rates of change before 2015 as the annualized change rate of indicators between 2015-2030 required by 25 SDG targets. Hence, targets which are not quantitatively defined can be specified by assuming their progress is keeping pace with defined indicators. We set quantitative targets for SDG3.9 based on the 10th percentile reduction in DAPP at the national scale during 2000-2015, and the estimated value is around 30%. In addition, to acknowledge the different levels of target setting and the uncertainty of pathways, following Moallemi et al. (2021)⁵, we roughly defined the substantial reduction in terms of three levels of ambition in DAPP reduction targets under SDG3.9, where weak, moderate, and ambitious targets are represented by a 10%, 20%, and 30% reduction compared to 2015 levels, respectively.”

Issue #2-2: Authors discuss adequately the limitations of this study. They should conduct sensitivity analyses when considering future projections in joint exposures to temperature and PM_{2.5} or they should at least discuss it.

Response #2-2: Revised as suggested.

Following the reviewer’s suggestion, we added the discussion on the joint exposure to temperature and PM_{2.5} in the **Discussion** section. The method used in this study cannot differentiate the joint effect of different risk factors (e.g., temperature and PM_{2.5} pollution). The revised version was shown in line 372-375 as follows:

“in estimating DAPP, although we used the latest method from GBD 2019⁶, the

epidemiological model cannot differentiate the interactive effects between PM_{2.5} pollution and other closely related risk factors (e.g., climate change-induced heatwaves, ozone pollution) and consideration of these effects is required in future analyses^{7,8}.”

Added references:

Li, K. *et al.* A two-pollutant strategy for improving ozone and particulate air quality in China. *Nature Geoscience* **12**, 906-910, doi:10.1038/s41561-019-0464-x (2019).

Patel, D. *et al.* Joint effects of heatwaves and air quality on ambulance services for vulnerable populations in Perth, western Australia. *Environmental Pollution* **252**, 532-542, doi:https://doi.org/10.1016/j.envpol.2019.05.125 (2019).

Issue #2-3: Please provide further rationale on why those 6 specific diseases related to air pollution were evaluated.

Response #2-3: Revised as suggested.

We have clarified this issue in the revised manuscript. In our study, we mainly used the methodologies of GBD 2019 and considered 6 diseases, which account for about 95% of total deaths related to PM_{2.5} pollution. Therefore, we have clarified in the **Methods** section. The revised version was shown in line 448-455 as follows:

“Six diseases related to PM_{2.5} pollution were considered in this study, including lung cancer, chronic obstructive pulmonary disease, lower respiratory infection, ischemic heart disease, stroke, and diabetes mellitus type 2. Evidence linking these diseases with exposure to ambient air pollution was judged to be consistent with a causal relationship on the basis of criteria specified for Global Burden of Disease (GBD) risk factors, including meta-analysis, cohort study, and biologically plausible relationship⁶. Although PM_{2.5} pollution is also related to other adverse birth outcomes including low birth weight and short gestation, these 6 diseases represent around 95% of total deaths related to PM_{2.5} pollution from all causes based on the estimation of GBD 2019⁶.”

Issue #2-4: Exposure-response functions were used and derived from published meta-regression models. This is appropriate. However, exposure-response functions do not appear to account for age-group specific associations as well as gender specific associations. Also, what about providing projections by age group and gender? Can authors clarify?

Response #2-4: Clarify and Revised.

As the reviewer mentioned, the exposure-response functions are age-specific for some

diseases. We further clarified the details of age groups in the **Methods** section. The revised version was shown in line 447-450 and line 465-468 as follows:

“In our formulation, 17 age groups were included in the equation, i.e., 15-20, 20-25, ..., 90-95, and 95+ years old. Six diseases related to PM_{2.5} pollution were considered in this study, including lung cancer, chronic obstructive pulmonary disease, lower respiratory infection, ischemic heart disease, stroke, and diabetes mellitus type 2.”

“For ischemic heart disease and stroke, we used age-specific exposure-response functions because the epidemiological evidence suggests that the relative risks for these diseases change by age^{9,10}. For the other four diseases, the exposure-response functions are uniform for all age groups.”

With regard to gender, the estimations of DAPP are also different by gender, because of their baseline death rates of diseases and their proportion. Because the availability of data, we did not consider the difference in gender in our projection. This issue was also clarified in the **Methods** section. The revised version was shown in line 455-456 as follows:

“We did not consider differences by gender to reduce the complexity of our projections.”

In addition, we added some age-specific results in the **Results** section and supplementary files. The revised version was shown in line 222-225 as follows:

“The share of DAPP amongst older people (65+) increased from almost 65% in 2015 to around 70% under all scenarios by 2030 due to the increase in older adults and higher life expectancy occurred alongside better healthcare (Figure S5).”

Fig. S5. Trends of deaths attributable to PM_{2.5} pollution by age group. The results were estimated based on multi-model averaged PM_{2.5} concentration, medium value of exposure-response function and death rate of disease. Please refers to Fig. S1 for detailed definitions of ASIA, OECD, MAF, REF and LAM.

Issue #2-5: Projections are conducted across large geographical areas. What about the importance of evaluating sub-national projections?

Response #2-5: Revised as suggested.

As the reviewer mentioned, sub-national specific results are important for local stakeholder and communities. Because of the availability of data (especially the death rates of specific

diseases), we were only able to project the DAPP at national scale. Sub-national results are indeed needed in the future. We emphasized this issue in **Discussion** section. The revised version was shown in line 392-397 as follows:

“Considering the availability of data, projections of death rate of diseases and age structure in this study were conducted at the national level, which cannot reflect the heterogeneity within countries which may be significant especially in large and populous countries (e.g., United States, China, India). Further projections at sub-national with finer resolution is needed to support decision-making of communities and stakeholders at regional and local scale^{11,12}.”

Issue #2-6: This study specifically focused on PM_{2.5} projections, but there could also be other pollutants affected such as ozone. Can authors comment on this issue?

Response #2-6: Revised as suggested.

We believe that, in addition to PM_{2.5}, other pollutants are also important. For example, ozone is an important air pollutant too. We added some explanation the **Introduction** to emphasize the importance of and ozone. The revised version was shown in line 75-76 as follows:

“Within the total deaths related to air pollution (generally considered PM_{2.5} and ozone), DAPP accounts for more than 95%^{13,14}.”

We also acknowledged the inadequate of our estimation that didn't consider other risk factors in the **Limitations** section. The revised version was shown in line 372-375 as follows:

“in estimating DAPP, although we used the latest method from GBD 2019⁶, the epidemiological model cannot differentiate the interactive effects between PM_{2.5} pollution and other closely related risk factors (e.g., climate change-induced heatwaves, ozone pollution) and consideration of these effects is required in future analyses^{7,8}.”

Added references:

Wang, Y. *et al.* Health Burden and economic impacts attributed to PM_{2.5} and O₃ in china from 2010 to 2050 under different representative concentration pathway scenarios. *Resources, Conservation and Recycling* **173**, 105731, doi:10.1016/j.resconrec.2021.105731 (2021).

Conibear, L. *et al.* The contribution of emission sources to the future air pollution disease burden in China. *Environmental Research Letters* **17**, 064027, doi:10.1088/1748-9326/ac6f6f (2022).

Li, K. *et al.* A two-pollutant strategy for improving ozone and particulate air quality in China. *Nature Geoscience* **12**, 906-910, doi:10.1038/s41561-019-0464-x (2019).

Patel, D. *et al.* Joint effects of heatwaves and air quality on ambulance services for vulnerable populations in Perth, western Australia. *Environmental Pollution* **252**, 532-542, doi:<https://doi.org/10.1016/j.envpol.2019.05.125> (2019).

Issue #2-7: Projections that were provided in this study looked specifically at long term effects of PM_{2.5}. There are also concerns regarding its short-term effects. Can this be incorporated into the analysis to obtain a better overall impact of the issue. Otherwise, please comment on this issue.

Response #2-7: Clarified and revised.

As the reviewer mentioned, our estimation used the epidemiological model developed by GBD2019 which based on long term effects, which did not consider the short-term effects. We acknowledged this inadequate in **Limitations** section. The revised version was shown in line 375-379 as follows:

“As our model reflected the long-term effects of PM_{2.5}, and estimated DAPP based on annual PM_{2.5} concentration, the short-term effects which associated with acute cardiovascular and respiratory diseases were not considered¹⁵. Further estimations of DAPP considering short-term epidemiological studies are needed in the future¹⁶.”

Added references:

Kloog, I., Ridgway, B., Koutrakis, P., Coull, B. A. & Schwartz, J. D. Long- and Short-Term Exposure to PM_{2.5} and Mortality: Using Novel Exposure Models. *Epidemiology* **24** (2013).

Liu, C. *et al.* Ambient Particulate Air Pollution and Daily Mortality in 652 Cities. *New England Journal of Medicine* **381**, 705-715, doi:10.1056/NEJMoa1817364 (2019).

Issue #2-8: Impacts of PM_{2.5} on health outcomes can be non-linear (or even supralinear) where the curve usually shows a steeper increase at lower concentrations and flattens out at higher concentrations. Please comment on how this would affect the future projections of PM_{2.5} on health.

Response #2-8: Clarified and Revised.

The exposure-response function is the core to translate PM_{2.5} pollution to attributable deaths, which shows a non-linear curve indeed. To illustrate this issue, we emphasized the shape of exposure-response curve in the **Methods** section. The revised version was shown in line 461-465 as follows:

“RR can be quantified based on a non-linear exposure-response function which usually shows

a steeper increase at lower concentrations with more modest increases at higher concentrations. In this study, we used the latest meta-regression (i.e., Bayesian, regularized, trimmed [MR-BRT] model) exposure-response functions updated by GBD 2019⁶ (Figure S3).”

Fig. S3. Exposure-response functions of Bayesian, regularized, trimmed (MR-BRT) model derived from GBD 2019 Risk Factors Collaborators (2020). IHD, COPD, LC and LRI refer to ischemic heart disease, chronic obstructive pulmonary disease, lung cancer, and lower respiratory infection, respectively. 35-40, 55-60 and 75-80 refers to the specific age groups. Shades refers to 95% confidence interval.

In addition, we also extended the **Results** section to illustrate the non-linear effects of exposure-response curve. The revised version was shown in line 235-241 as follows:

“As the relationship between $PM_{2.5}$ concentration and health effects represented by exposure-response curves is non-linear, DAPP tended to show a steeper increase at lower concentrations and levelled off at higher concentrations (Figure S3). Hence, a lower level of $PM_{2.5}$ concentration in the future would substantially reduce DAPP. For example, under SSP1-2.6 a decrease in $PM_{2.5}$ concentration of 31% led to a 24% decrease in DAPP from 2015-2030; whereas under SSP5-8.5, a growth of 12% in $PM_{2.5}$ concentration led to a 2% increase in DAPP (Figure 5).”

Issue #2-9: Line 91 : I believe it should read “and the results are limited in capturing the range...” , Line 121: This part if repeated twice: “a great challenge”.

Response #2-9: Revised as suggested.

We removed the unwanted words and edited the entire manuscript for brevity and clarity.

Response to the reviewer 3

The study analyses the potential reduction of premature mortalities attributable to ambient PM_{2.5} (DAPP) under alternative SSP-RCP scenarios, and if/how do they align with the “substantial” reduction mentioned by the UN’s SDG 3.9. The manuscript studies the influences of four key drivers (PM_{2.5} concentration levels, baseline mortality rates of diseases, population growth, and aging) for the evolution of future DAPPs for different regions over the world. The topic is of interest for the scientific community and for non-expert stakeholders, as it shows the need for more ambitious action to obtain a significant reduction in premature mortalities associated with air pollution. I have some general comments.

Issue #3-1: First, I think the contribution of the study could be reframed. There exist several global and regional studies that assess the reduction of DAPPs for different scenarios which already consider dynamic PM_{2.5} concentration levels, population structure and aging, or baseline mortality rates. Moreover, these studies include a wide range of scenarios that go beyond the standard SSP-RCP scenarios from ScenarioMIP.

Response #3-1: Accepted and revised.

As the reviewer mentioned, there are other studies beyond standard SSP-RCP scenarios. We reframed our contribution in **Discussion**. The revised version was shown in line 285-307 as follows:

“Many studies have assessed the trends in DAPP including dynamic PM_{2.5} concentration levels, population structure and aging, and death rates of diseases based on various scenarios (including SSP-RCP scenarios) (Table S1). However, a lack of internally consistent, comprehensive scenario analyses has led to wide variations in future projections of DAPP which differ not only in magnitude but also in sign, potentially resulting in divergent interpretations and policy responses. As a response, a major contribution of this study is providing a comprehensive long-term projection of DAPP with all 11 available simulations from CMIP6 under the latest ScenarioMIP framework, which provides ensembles of integrated, internally consistent estimates for the drivers of DAPP (population, age structure, death rate of diseases, and air quality)². Thus, our projections avoid unintended logical conflicts caused by using independent and internally inconsistent assumptions around driving factors¹⁷ and improve the accuracy and robustness of projections. Compared with another study¹⁸ that did project DAPP using coherent and internally consistent assumptions for each relevant factor under the ScenarioMIP framework with PM_{2.5} concentration simulated by GFDL-ESM4.1, our study also provides greater robustness by including all available CMIP6

general circulation models. In addition, estimation based on a well-established scenario framework also makes our results more comparable with other studies under the SSPs and CMIP6 framework, which can support integrated decision-making by jointly considering the results from multiple fields^{5,19}.

This work is also the first attempt to assess potential future progress and development pathways towards SDG3.9 and does so on multiple scales: global, regional, national, and by grid cell at a spatial resolution of 1 degree. The results shed light on guiding future air pollution control policy and healthcare improvement and provide the robust projections required to support the management of health impacts caused by PM_{2.5} pollution for the scientific community and stakeholders.”

We also summarized some latest research in Table S1 which include 13 references and extended the review of previous studies.

Table S1. Comparative summary of scenario, input data and major results among previous studies.

Reference	Source	Study area	PM _{2.5} concentration projection		Socioeconomic factors	Changes in DAPP (deaths attributable to PM _{2.5} pollution)
			Model	Scenario		
Huang et al. (2023) ²⁰	Nature Sustainability	World	TM5-FASST	30000 state of worlds sampled by considering five set of SSPs and carbon price.	Population, age structure and baseline death rate in line with SSPs.	With pricing carbon, ensemble-median DAPP decreased 0.5 million annually from 2015 to 2100 on average.
Cheng et al (2023) ²¹	One Earth	China	WRF-CMAQ	Specific scenarios defined by clean air and carbon neutrality policy.	Population, age structure in line with SSP1 and baseline death rate change by time.	DAPP is expected to decrease from 1.36 in 2020 to 1.28 in 2030 via the combination of carbon-peak, carbon-neutrality, and air pollution control policies
Yang et al. (2022) ²⁶	Nature Sustainability	World	GFDL-ESM4.1	SSP1-1.9, SSP1-2.6, SSP2-4.5, SSP3-7.0 and SSP5-8.5	Population, age structure and baseline death rate in line with SSPs.	DAPP is expected to increase by 7.3% (SSP1-1.9) -54.9% (SSP5-8.5) from 2015 to 2030.
Sliva et al. (2017) ¹⁸	Nature Climate Change	World	5 models from CMIP5	RCP8.5, assuming the emissions fixed	Population, age structure and baseline death rate change by time.	DAPP is expected to increase by 3.3% from 2000 to 2030.
Hong et al. (2019) ²⁷	Proceedings of the National Academy of Sciences	China	WRF-CMAQ	RCP4.5, assuming the emissions fixed	Population and baseline death rate change by time while assuming fixed age structure.	DAPP is expected to increase by 27 thousand from 2006-2010 to 2046-2050.
Markandya et al. (2009) ²⁸	Lancet	Europe, India and China	TM5-FASST	Business as usual scenario, assuming the meteorological field fixed	Baseline death rate change over time, did not differentiate by age structure.	DAPP per million people is expected to decrease by 41% and 4% in Europe and China, while increasing by 38% in India from 2010 to 2030.
Lelieveld et al. (2015) ¹⁴	Nature	World	ECHAM5	Business as usual scenario, assuming the meteorological field fixed	Population and age structure change over time while assuming fixed baseline death rate.	DAPP is expected to increase by 97% from 2010 to 2050.
Fang et al. (2013) ²⁹	Climatic Change	World	GFDL-AM3	SRES-A1B	Population, age structure and baseline death rate fixed.	DAPP is expected to increase by 4.4% from 2000 to 2090.
Rafaj et al. (2018) ³⁰	Global Environmental Change	World	EMEP MSC-W	3 energy-related scenarios based on World Energy Model, assuming the meteorological field fixed	Population and age structure change over time while assuming contributions from individual diseases to total deaths within each age group remain constant.	DAPP is expected to increase by 40% from 2015 to 2040, under the scenario assuming the continuation of existing and planned policies.
West et al. (2013) ³¹	Nature Climate Change	World	MOZART-4	RCP4.5	Population and baseline death rate change over time, did not differentiate age structure.	DAPP is expected to increase by 0.5 million in 2030, but decrease by 2.4 million in 2050 relative to 2000.
Sliva et al. (2016) ³²	Atmospheric Chemistry and Physics	World	6 models from CMIP5	RCP2.6, RCP4.5, RCP6.0, and RCP8.5	Population, age structure and baseline death rate change over time.	DAPP is expected to increase from 1.7 million in 2000 to 2.5 million in 2030, and then drop to 1.8 million in 2050 under RCP4.5.
Chowdhury et al. (2018) ¹⁹	Nature Communications	India	13 models from CMIP5	RCP4.5 and RCP8.5	Population, age structure and baseline death rate change by time.	DAPP is expected to decrease by 12% (RCP4.5) from 2011-2020 to 2031-2040.
Yang et al. (2019) ³³	Environmental Research	United States	WRF-Chem	RCP4.5 and RCP8.5	Population and baseline death rate change over time, did not differentiate age structure.	DAPP is expected to decrease by 63 thousand (RCP4.5) and 82 thousand (RCP8.5) in 2030, relative to 2005.

Issue #3-2: Likewise, the study also explores the effects of additionally reducing PM_{2.5} levels and improving the healthcare system by 20%, an arbitrary magnitude that drives to an expected or obvious result.

Response #3-2: Clarified and revised.

As the reviewer mentioned, the magnitude of 20% was set subjectively in the sensitivity analysis. It's intuitive that with this extra effort, the attainment of SDG3.9 will be easier in the future. However, we believe that it's still useful to quantify the attainment of SDG3.9 under different assumptions because the non-linear relationship between DAPP and the changes in PM_{2.5} concentration and death rate of diseases. We clarified this issue in **Methods**. The revised version was shown in line 549-561 as follows:

“Based on ScenarioMIP projection, we also considered the potential of additional efforts to reduce DAPP. Among the four components influencing DAPP, demographic factors (population and age structure) cannot be effectively altered by policy intervention in the short term, but PM_{2.5} concentration and death rate of diseases can more readily be changed via additional efforts in air pollution control (e.g., stricter air quality standards, technological innovation, and cleaner energy mix)^{20,21} and healthcare (e.g., investment in medical research, scientific breakthroughs, health system improvements, and better accessibility)^{22,23}. We considered three possible conditions, including additional air quality improvement (20% lower PM_{2.5} concentration + No change in death rate of diseases); additional healthcare improvement (No change in PM_{2.5} concentration + 20% lower death rate of diseases) and both measures (20% lower PM_{2.5} concentration + 20% lower death rate of diseases). The value of 20% was selected subjectively to represent a significant and plausible improvement resulting from concerted government attention. With these assumptions, we conducted a sensitivity analysis to assess the response of additional air quality and healthcare interventions towards SDG3.9.”

Added references:

Cheng, J. *et al.* A synergistic approach to air pollution control and carbon neutrality in China can avoid millions of premature deaths annually by 2060. *One Earth* **6**, 978-989, doi:10.1016/j.oneear.2023.07.007 (2023).

Issue #3-3: On the other hand, the study provides same interesting results, such as the gridded or city-level analysis of the future reduction in DAPPs, or the decomposition analysis. I believe these should be the main focus of the paper in order to make it more relevant for the community.

Response #3-3: Accept and revised.

As the reviewer suggested, we reframed **Effects of individual factors on the attainment of SDG3.9** section in **Results**, to emphasize the finding from decomposition analysis. The revised version was shown in line 216-251 as follows:

“Population aging was the most significant factor under all scenarios in driving growth in DAPP, causing an increase of 34% (SSP3-7.0) to 41% (SSP5-8.5) by 2030. Conversely, healthcare improvement was the strongest factor driving down DAPP, leading to a decline of 36% (SSP3-7.0) to 44% (SSP5-8.5). As socioeconomic development reduced the death rate of diseases and exacerbated population aging simultaneously, these two drivers tended to offset each other, resulting in a relatively minor net decrease of around 1 to 3% (Figure 5). This counteracting effect also influenced the age distribution of DAPP. The share of DAPP amongst older people (65+) increased from almost 65% in 2015 to around 70% under all scenarios by 2030 due to the increase in older adults and higher life expectancy occurred alongside better healthcare (Figure S5). Population growth led to a minor increase in DAPP of around 6% (SSP1-2.6) to 14% (SSP3-7.0).

Compared with other driving factors, the effect of PM_{2.5} concentration was more variable across scenarios. For instance, DAPP in 2030 was projected to decrease by 24% relative to 2015 because of the decline in PM_{2.5} concentration alone under SSP1-2.6, led by the effort in climate mitigation and strong air pollution control. Similarly, air quality improvement was responsible for a decline of 15% and 8% in DAPP under SSP5-8.5 (strong air pollution control and high energy demand) and SSP2-4.5 (moderate air pollution control and moderate energy demand), respectively. Under the worst scenario for air pollution (SSP3-7.0) which assumes a weak air pollution control with slow socioeconomic development, DAPP was projected to increase by 2% due to a 12% increase in PM_{2.5} concentration relative to 2015 (Figure 5; Figure S15). As the relationship between PM_{2.5} concentration and health effects represented by exposure-response curves is non-linear, DAPP tended to show a steeper increase at lower concentrations and levelled off at higher concentrations (Figure S3). Hence, a lower level of PM_{2.5} concentration in the future would substantially reduce DAPP. For example, under SSP1-2.6 a decrease in PM_{2.5} concentration of 31% led to a 24% decrease in DAPP from 2015-2030; whereas under SSP5-8.5, a growth of 12% in PM_{2.5} concentration led to a 2% increase in DAPP (Figure 5).

Similar patterns in driving factors occurred beyond 2030 at the global scale. Aging and healthcare improvement remained the dominant driving factors affecting DAPP and the attainment of SDG3.9 while the effect of PM_{2.5} concentration varied among scenarios. The effect of population growth between 2030-2050 was smaller relative to that from 2015-2030

because population growth was slower. Detailed region- and country-specific decomposition were also conducted (Figure S19-20). For most countries, changes in aging and healthcare were the main contributors to DAPP.”

We then emphasized the interactions among factors in **Discussion** section. The revised version was shown in line 309-324 as follows:

“Global DAPP will not be substantially reduced under all but the most ambitious scenarios (SSP1-2.6) because of the overwhelming effect of population aging (Figure 1). However, there are complex interactions between different scenarios. For instance, with socioeconomic development, the population will get older which increases the size of vulnerable populations, but the death rate of diseases declines which simultaneously reduces population vulnerability. Hence, the effect of population aging will be offset by improvement in healthcare, hence scenarios with high population aging (SSP1-2.6 and SSP5-8.5) showed greater reduction in DAPP. Besides the improvement in healthcare, changes in PM_{2.5} concentration are also a key driver of reducing DAPP. SSP1-2.6 had the largest reduction because of the underlying assumption of a cleaner energy mix and strong air pollution control. However, PM_{2.5} concentration is also expected to reduce in SSP5-8.5. That is because of the strong air pollution control led by the adoption of the current best available technology (especially end-of-pipe control) and socio-economic development based on high energy demand and fossil fuel dominated energy mix. Hence, transformation aligned with an SSP1-2.6 future with stronger air quality control and less fossil fuel dependence is the better pathway for reducing DAPP (Figure 5, Figure S9) as it has positive co-benefits for many other SDGs that are not assessed in this study.”

We also made regional and national specific interpretation to make our paper more relevant for community and stakeholders. The revised version was shown in line 325-333 as follows:

“In addition, the interaction between different drivers varied by region. For instance, besides population aging, population growth in MAF also played a considerable role under all scenarios because the region’s population is relatively young and growing fast. While in LAM and ASIA, aging is the dominant driver of growth in DAPP (Figure S19). Different countries also exhibited heterogeneity in mechanisms driving DAPP. For instance, China is likely to achieve SDG3.9 under most scenarios because of the reversal of population growth and the declining trend of PM_{2.5} pollution, as well as healthcare improvement. While for India, DAPP tended to increase under three scenarios (all but SSP1-2.6) mainly because of the joint effect of air quality deterioration and demographic change.”

Issue #3-4: I also have some concerns related to some of the assumptions that should be

further explained. First, the study assumes that investing in healthcare would directly reduce the baseline mortality rates, which directly decreases DAPPs. This is important since it seems to be the strongest factor for decline of DAPPs (Line 178). This causality is not very obvious and I think it should be further elaborated considering its implications.

Response #3-4: Clarified and revised.

As the reviewer mentioned, the relationship between investing in healthcare and death rate of diseases is not clear. In our study, we assumed a 20% decrease of death rate of diseases, which might be caused by multiple factors, including increases in investment in medical research, scientific breakthroughs, health system improvements and better accessibility. We clarified this in the **Methods** section. The revised version was shown in line 550-560 as follows:

“Among the four components influencing DAPP, demographic factors (population and age structure) cannot be effectively altered by policy intervention in the short term, but PM_{2.5} concentration and death rate of diseases can more readily be changed via additional efforts in air pollution control (e.g., stricter air quality standards, technological innovation, and cleaner energy mix)^{20,21} and healthcare (e.g., investment in medical research, scientific breakthroughs, health system improvements, and better accessibility)^{22,23}. We considered three possible conditions, including additional air quality improvement (20% lower PM_{2.5} concentration + No change in death rate of diseases); additional healthcare improvement (No change in PM_{2.5} concentration + 20% lower death rate of diseases) and both measures (20% lower PM_{2.5} concentration + 20% lower death rate of diseases). The value of 20% was selected subjectively to represent a significant and plausible improvement resulting from concerted government attention.”

Issue #3-5: Furthermore, Table 1 shows that the emission controls in the SSP5-8.5 scenario (development of FF) are similar to the controls in the Sustainability scenario (SSP1-2.6). Is this an assumption on ScenarioMIP? If not, this can be counterintuitive (and incorrect) and it should be explained in more detail.

Response #3-5: Clarified and revised.

This is the assumption of SSP5-8.5 scenario. In the underlying assumption, the air pollution control is closely related to socioeconomic development. In both SSP1-2.6 and SSP5-8.5 scenarios, their socioeconomic development are highly similar. We clarified this issue in **Scenario framework** section in **Methods**. The revised version was shown in line 432-435 as follows:

“Under this framework, studies have projected the corresponding socioeconomic factors and concentration of pollutants^{3,24,25}, with stricter air pollution controls tied to higher levels of economic development¹. Hence, weak air pollution controls occur in SSP3-7.0, with medium controls in SSP2-4.5, and strong air pollution controls in SSP1-2.6 and SSP5-8.5²”

Issue #3-6: Finally, regarding the presentation of the results, I believe that description of some of the results is too vague (particularly in the first results subsection) and too descriptive without getting deeper into the reasons or drivers. In addition, the discussion is quite repetitive. My recommendation for the discussion would be to take the first paragraphs out, and mix the points raised from line 288 with the limitations (line 325). Overall, the study comprises some interesting information that would be relevant for the community, but I believe it should be substantially reframed, in order to focus on those aspects (see above) that would be the most relevant for the scientific community and expert and non-expert stakeholders.

Response #3-6: Accepted and revised.

At first, we shorten the first **Results** subsection to avoid vague and descriptive expression, and pay more attention to deeper driver. The revised version was shown in line 134-164 as follows:

“The multi-model average results indicate that global DAPP was substantially reduced only under those scenarios with the most ambitious assumptions around continuing growth and aging in the global population and declines in death rates of diseases (Figure S15). Thus, achieving SDG3.9 remains a great challenge. The SSP1-2.6 scenario saw the largest decrease, with average DAPP projected to almost meet the moderate target (i.e., a 20% reduction compared to 2015 levels) by 2030 (-19%) and exceed the target by 2050 (-24%). Under all other scenarios, the moderate target of SDG3.9 was not achieved. Average DAPP also declined (-11% by 2030 and -7% by 2050) under the SSP5-8.5 scenario, but remained stable under the SSP2-4.5 scenario (-1% by 2030 and +2% by 2050). In the worst case, average DAPP grew 14% by 2030 and 32% by 2050 under SSP3-7.0 (Figure 2a). Amongst different age groups, the share of DAPP for older people (65+) accounted for almost 65% in 2015 and rose to 70% under all scenarios by 2030 because older adults have a higher baseline death rate and are more vulnerable to almost all types of health risk (Figure S5).

Middle East and Africa (MAF) and Latin America and the Caribbean (LAM) were hotspots of future growth in DAPP, increasing between 11% (SSP1-2.6) and 101% (SSP3-7.0) by 2030, and between 13% (SSP1-2.6) and 105% (SSP3-7.0) by 2050, respectively. SDG3.9 achievement in these regions is not expected in the future (Figure 2c, e). On the contrary,

DAPP in the member states of the Organization for Economic Co-operation and Development and new European Union and candidates (OECD) and the reforming economies of Eastern Europe and the Former Soviet Union (REF) tended to decrease under most scenarios. The OECD met the moderate SDG3.9 target under SSP1-2.6 by 2030 and 2050 while REF even achieved the ambitious target (30% reduction) (Figure 2d, f).”

Secondly, we also add more interpretation in the second subsection in **Results** to present the deeper mechanism and driver. The revised version was shown in line 216-251 as follows:

“Population aging was the most significant factor under all scenarios in driving growth in DAPP, causing an increase of 34% (SSP3-7.0) to 41% (SSP5-8.5) by 2030. Conversely, healthcare improvement was the strongest factor driving down DAPP, leading to a decline of 36% (SSP3-7.0) to 44% (SSP5-8.5). As socioeconomic development reduced the death rate of diseases and exacerbated population aging simultaneously, these two drivers tended to offset each other, resulting in a relatively minor net decrease of around 1 to 3% (Figure 5). This counteracting effect also influenced the age distribution of DAPP. The share of DAPP amongst older people (65+) increased from almost 65% in 2015 to around 70% under all scenarios by 2030 due to the increase in older adults and higher life expectancy occurred alongside better healthcare (Figure S5). Population growth led to a minor increase in DAPP of around 6% (SSP1-2.6) to 14% (SSP3-7.0).

Compared with other driving factors, the effect of PM_{2.5} concentration was more variable across scenarios. For instance, DAPP in 2030 was projected to decrease by 24% relative to 2015 because of the decline in PM_{2.5} concentration alone under SSP1-2.6, led by the effort in climate mitigation and strong air pollution control. Similarly, air quality improvement was responsible for a decline of 15% and 8% in DAPP under SSP5-8.5 (strong air pollution control and high energy demand) and SSP2-4.5 (moderate air pollution control and moderate energy demand), respectively. Under the worst scenario for air pollution (SSP3-7.0) which assumes a weak air pollution control with slow socioeconomic development, DAPP was projected to increase by 2% due to a 12% increase in PM_{2.5} concentration relative to 2015 (Figure 5; Figure S15). As the relationship between PM_{2.5} concentration and health effects represented by exposure-response curves is non-linear, DAPP tended to show a steeper increase at lower concentrations and levelled off at higher concentrations (Figure S3). Hence, a lower level of PM_{2.5} concentration in the future would substantially reduce DAPP. For example, under SSP1-2.6 a decrease in PM_{2.5} concentration of 31% led to a 24% decrease in DAPP from 2015-2030; whereas under SSP5-8.5, a growth of 12% in PM_{2.5} concentration led to a 2% increase in DAPP (Figure 5).

Similar patterns in driving factors occurred beyond 2030 at the global scale. Aging and

healthcare improvement remained the dominant driving factors affecting DAPP and the attainment of SDG3.9 while the effect of PM_{2.5} concentration varied among scenarios. The effect of population growth between 2030-2050 was smaller relative to that from 2015-2030 because population growth was slower. Detailed region- and country-specific decomposition were also conducted (Figure S19-20). For most countries, changes in aging and healthcare were the main contributors to DAPP.”

Besides, we also deleted the first paragraphs in **Discussion** section to avoid ambiguity and repetition, and simplified the part of **Policy, innovation, and investment for achieving SDG3.9** which used in line 228. The revised version was shown in line 365-370 as follows:

“Stronger efforts in air pollution control and healthcare are also co-beneficial for achieving several other SDGs. Efforts in air pollution control are in line with mitigation and adaptation commitments in SDG13 (Climate Action)^{26,27} and promote progress toward many other aspects of sustainability. Hence, comprehensive policies that team air pollution control and public health with climate change mitigation efforts, technological innovation, and energy system overhaul could help meet multiple SDGs²⁸⁻³⁰.”

Issue #3-7: (Line 78) I think there is a substantial body of literature exploring AAP-related health impacts, also for SSP and decarbonization scenarios. The comparison should include more recent studies that have not been considered.

Response #3-7: Accepted and revised.

As the reviewer suggested, we added more recent studies in our background and comparison with previous studies and made modification throughout the revised paper. The relevant information from 13 papers was summarized in Table S1 as follow:

Table S1. Comparative summary of scenario, input data and major results among previous studies.

Reference	Source	Study area	PM _{2.5} concentration projection		Socioeconomic factors	Changes in DAPP (deaths attributable to PM _{2.5} pollution)
			Model	Scenario		
Huang et al. (2023) ²⁰	Nature Sustainability	World	TM5-FASST	30000 state of worlds sampled by considering five set of SSPs and carbon price.	Population, age structure and baseline death rate in line with SSPs.	With pricing carbon, ensemble-median DAPP decreased 0.5 million annually from 2015 to 2100 on average.
Cheng et al (2023) ²¹	One Earth	China	WRF-CMAQ	Specific scenarios defined by clean air and carbon neutrality policy.	Population, age structure in line with SSP1 and baseline death rate change by time.	DAPP is expected to decrease from 1.36 in 2020 to 1.28 in 2030 via the combination of carbon-peak, carbon-neutrality, and air pollution control policies
Yang et al. (2022) ²⁶	Nature Sustainability	World	GFDL-ESM4.1	SSP1-1.9, SSP1-2.6, SSP2-4.5, SSP3-7.0 and SSP5-8.5	Population, age structure and baseline death rate in line with SSPs.	DAPP is expected to increase by 7.3% (SSP1-1.9) -54.9% (SSP5-8.5) from 2015 to 2030.
Sliva et al. (2017) ¹⁸	Nature Climate Change	World	5 models from CMIP5	RCP8.5, assuming the emissions fixed	Population, age structure and baseline death rate change by time.	DAPP is expected to increase by 3.3% from 2000 to 2030.
Hong et al. (2019) ²⁷	Proceedings of the National Academy of Sciences	China	WRF-CMAQ	RCP4.5, assuming the emissions fixed	Population and baseline death rate change by time while assuming fixed age structure.	DAPP is expected to increase by 27 thousand from 2006-2010 to 2046-2050.
Markandya et al. (2009) ²⁸	Lancet	Europe, India and China	TM5-FASST	Business as usual scenario, assuming the meteorological field fixed	Baseline death rate change over time, did not differentiate by age structure.	DAPP per million people is expected to decrease by 41% and 4% in Europe and China, while increasing by 38% in India from 2010 to 2030.
Lelieveld et al. (2015) ¹⁴	Nature	World	ECHAM5	Business as usual scenario, assuming the meteorological field fixed	Population and age structure change over time while assuming fixed baseline death rate.	DAPP is expected to increase by 97% from 2010 to 2050.
Fang et al. (2013) ²⁹	Climatic Change	World	GFDL-AM3	SRES-A1B	Population, age structure and baseline death rate fixed.	DAPP is expected to increase by 4.4% from 2000 to 2090.
Rafaj et al. (2018) ³⁰	Global Environmental Change	World	EMEP MSC-W	3 energy-related scenarios based on World Energy Model, assuming the meteorological field fixed	Population and age structure change over time while assuming contributions from individual diseases to total deaths within each age group remain constant.	DAPP is expected to increase by 40% from 2015 to 2040, under the scenario assuming the continuation of existing and planned policies.
West et al. (2013) ³¹	Nature Climate Change	World	MOZART-4	RCP4.5	Population and baseline death rate change over time, did not differentiate age structure.	DAPP is expected to increase by 0.5 million in 2030, but decrease by 2.4 million in 2050 relative to 2000.
Sliva et al. (2016) ³²	Atmospheric Chemistry and Physics	World	6 models from CMIP5	RCP2.6, RCP4.5, RCP6.0, and RCP8.5	Population, age structure and baseline death rate change over time.	DAPP is expected to increase from 1.7 million in 2000 to 2.5 million in 2030, and then drop to 1.8 million in 2050 under RCP4.5.
Chowdhury et al. (2018) ¹⁹	Nature Communications	India	13 models from CMIP5	RCP4.5 and RCP8.5	Population, age structure and baseline death rate change by time.	DAPP is expected to decrease by 12% (RCP4.5) from 2011-2020 to 2031-2040.
Yang et al. (2019) ³³	Environmental Research	United States	WRF-Chem	RCP4.5 and RCP8.5	Population and baseline death rate change over time, did not differentiate age structure.	DAPP is expected to decrease by 63 thousand (RCP4.5) and 82 thousand (RCP8.5) in 2030, relative to 2005.

Issue #3-8: (Line 99) The study explores SSP-RCP combinations, not just SSP narratives.

Response #3-8: Accepted and revised.

The scenarios used in this study were designed by ScenarioMIP, which represent a matrix of possible integration of multiple SSPs and forcing outcomes. According to the original descriptions of ScenarioMIP (O'Neill et al., 2016), the scenarios used in this study were driven by a set of emissions and land use scenarios (Riahi et al., 2016), produced with integrated assessment models (IAMs) based on new future pathways of societal development, the Shared Socioeconomic Pathways (SSPs). Hence, we replaced the “SSPs scenarios” with “SSPs based scenarios” throughout the manuscript. We didn't use SSP-RCP combined scenarios directly because this might mean combination of socioeconomic factors under SSPs and PM_{2.5} simulations from CMIP5 under RCPs in some studies.

We also clarified this issue in **Scenario framework** section in **Methods**. The revised version was shown in line 416-423 as follows:

“ScenarioMIP is a new, widely accepted scenario framework^{31,32} which provides integrated, internally consistent ensemble simulations of driving factors suitable for research across multiple scientific fields (Table 1). ScenarioMIP represents a matrix of possible integration of multiple SSPs and forcing outcomes driven by a set of emissions and land use scenarios, produced with integrated assessment models (IAMs)³³. These scenarios were represented as SSP x–y, where x is the societal conditions described by SSPs and y represents the forcing pathway that could be made consistent with SSP emissions³⁴. Compared with previous RCPs derived from earlier emissions and land use scenarios³⁵, ScenarioMIP scenarios provided a related but updated simulation.”

Added reference:

O'Neill, B. C. *et al.* Achievements and needs for the climate change scenario framework. *Nature Climate Change* **10**, 1074-1084, doi:10.1038/s41558-020-00952-0 (2020).

van Vuuren, D. P. *et al.* The representative concentration pathways: an overview. *Climatic Change* **109**, 5, doi:10.1007/s10584-011-0148-z (2011).

Issue #3-9: Duplicated “a great challenge” .

Response #3-9: Accepted and revised. We have deleted the duplicated words.

Issue #3-10: - (Line 157) I think the information in Figure 3 is very powerful, but the format

of the figure is confusing... I would try to find another way of presenting these (interesting) results.

Response #3-10: Accepted and revised. We redesigned figure 3 in a more intuitive way to present the possibility of countries that can meet SDG3.9 under different scenarios. The revised version was shown in line 191-196 as follows:

Figure 4. Attainment of SDG3.9 by 2030 and 2050 for the 154 nations. The colors indicate percentages of countries that can meet SDG3.9 under 0, 1, 3, 2, and 4 possible scenarios assessed with, weak, moderate and ambitious settings for SDG3.9 represented as 10%, 20% and 30% reduction in DAPP relative to 2015, respectively. Only 8 models that were available for all the 4 scenarios were included in the model-specific analysis.

Issue #3-11: (Line 201) Again, I am not sure that investment in healthcare reduces the disease rates, at least in some regions.

Response #3-11: Clarified and revised.

As the reviewer mentioned, besides the investment in healthcare, the changes in death rate of diseases are influenced by multiple complex factors, the causality between investment healthcare and decrease in death rate of diseases is not clear. In our study, we assumed a 20% decrease of death rate of diseases, which might be caused by multiple factors, including increases in investment in medical research, scientific breakthroughs, health system improvements and better accessibility. We clarified this in the **Methods** section. The revised version was shown in line 550-560 as follows:

“Among the four components influencing DAPP, demographic factors (population and age structure) cannot be effectively altered by policy intervention in the short term, but PM_{2.5}

concentration and death rate of diseases can more readily be changed via additional efforts in air pollution control (e.g., stricter air quality standards, technological innovation, and cleaner energy mix)^{20,21} and healthcare (e.g., investment in medical research, scientific breakthroughs, health system improvements, and better accessibility)^{22,23}. We considered three possible conditions, including additional air quality improvement (20% lower PM_{2.5} concentration + No change in death rate of diseases); additional healthcare improvement (No change in PM_{2.5} concentration + 20% lower death rate of diseases) and both measures (20% lower PM_{2.5} concentration + 20% lower death rate of diseases). The value of 20% was selected subjectively to represent a significant and plausible improvement resulting from concerted government attention.”

Issue #3-12: (Line 227). As indicated, it is strange that, with no further justification, the Fossil-fuel-dependent scenario reduces more the PM_{2.5} concentration levels than other scenarios due to stringent air pollution control.

Response #3-12: Accepted and revised.

In the underlying assumption of SSP5-8.5, the air pollution control is closely related to socioeconomic development. Hence, the Fossil-fuel-dependent scenario have a strict air pollution control because of the socioeconomic development driven by fossil fuel. We clarified this issue in **Scenario framework** section in **Methods**. The revised version was shown in line 429-435 as follows:

“...Fossil-fueled Development (SSP5-8.5), representing a future with prosperous socioeconomic development embodied with improved air quality control at the expense of climate, with high energy demand¹.

Under this framework, studies have projected the corresponding socioeconomic factors and concentration of pollutants^{3,24,25}, with stricter air pollution controls tied to higher levels of economic development¹. Hence, weak air pollution controls occur in SSP3-7.0, with medium controls in SSP2-4.5, and strong air pollution controls in SSP1-2.6 and SSP5-8.5².”

Issue #3-13: (Line 313). It is quite obvious that more stringent controls and improving healthcare will reduce PM_{2.5}-related mortalities... I think this discussion is vague, and should consider other aspects such as costs, technological feasibility etc.

Response #3-13: Accepted and revised.

We simplified the part of interactions of SDGs which used in line 313 in last version. The

revised version was shown in line 365-370 as follows:

“Stronger efforts in air pollution control and healthcare are also co-beneficial for achieving several other SDGs. Efforts in air pollution control are in line with mitigation and adaptation commitments in SDG13 (Climate Action)^{26,27} and promote progress toward many other aspects of sustainability. Hence, comprehensive policies that team air pollution control and public health with climate change mitigation efforts, technological innovation, and energy system overhaul could help meet multiple SDGs²⁸⁻³⁰.”

We also emphasized the aspects such as costs, technological feasibility in **Policy, innovation, and investment for achieving SDG3.9** section in **Discussions**. The revised version was shown in line 362-364 as follows:

“Cost-effectiveness and technological feasibility should also be considered. Strategies like pricing carbon and pollutant emissions using market-based policy could be a feasible way to help developing countries to encourage innovation²⁰.”

Issue #3-14: (Line 357): It is not clear the distinction between SSP narratives and RCPs.

Response #3-14: Accepted and revised.

We clarified their relationship in **Scenario framework** section in **Methods**. The revised version was shown in line 418-421 as follows:

“ScenarioMIP represents a matrix of possible integration of multiple SSPs and forcing outcomes driven by a set of emissions and land use scenarios, produced with integrated assessment models (IAMs)³³. These scenarios were represented as SSP x–y, where x is the societal conditions described by SSPs and y represents the forcing pathway that could be made consistent with SSP emissions³⁴.”

Issue #3-15: (Line 463): Not additional explanation why 20% has been chosen.

Response #3-15: Clarified and revised.

As the reviewer mentioned, the magnitude of 20% was set subjectively in the sensitivity analysis. It's intuitive that with this extra effort, the attainment of SDG3.9 will be easier in the future. However, we believe that it's still useful to quantify the attainment of SDG3.9 under different assumptions because the non-linear relationship between DAPP and the changes in PM_{2.5} concentration and death rate of diseases. We clarified this issue in **Methods**. The revised version was shown in line 549-561 as follows:

“Based on ScenarioMIP projection, we also considered the potential of additional efforts to

reduce DAPP. Among the four components influencing DAPP, demographic factors (population and age structure) cannot be effectively altered by policy intervention in the short term, but PM_{2.5} concentration and death rate of diseases can more readily be changed via additional efforts in air pollution control (e.g., stricter air quality standards, technological innovation, and cleaner energy mix)^{20,21} and healthcare (e.g., investment in medical research, scientific breakthroughs, health system improvements, and better accessibility)^{22,23}. We considered three possible conditions, including additional air quality improvement (20% lower PM_{2.5} concentration + No change in death rate of diseases); additional healthcare improvement (No change in PM_{2.5} concentration + 20% lower death rate of diseases) and both measures (20% lower PM_{2.5} concentration + 20% lower death rate of diseases). The value of 20% was selected subjectively to represent a significant and plausible improvement resulting from concerted government attention. With these assumptions, we conducted a sensitivity analysis to assess the response of additional air quality and healthcare interventions towards SDG3.9.”

References:

- 1 Rao, S. *et al.* Future air pollution in the Shared Socio-economic Pathways. *Global Environmental Change* **42**, 346-358, doi:<https://doi.org/10.1016/j.gloenvcha.2016.05.012> (2017).
- 2 Gidden, M. J. *et al.* Global emissions pathways under different socioeconomic scenarios for use in CMIP6: a dataset of harmonized emissions trajectories through the end of the century. *Geoscientific Model Development* **2018**, 1-42, doi:10.5194/gmd-2018-266 (2018).
- 3 Turnock, S. T. *et al.* Historical and future changes in air pollutants from CMIP6 models. *Atmospheric Chemistry and Physics* **2020**, 1-40, doi:10.5194/acp-2019-1211 (2020).
- 4 GBD 2017 SDG Collaborators. Measuring progress from 1990 to 2017 and projecting attainment to 2030 of the health-related Sustainable Development Goals for 195 countries and territories: a systematic analysis for the Global Burden of Disease Study 2017. *The Lancet* **392**, 2091-2138, doi:10.1016/s0140-6736(18)32281-5 (2018).
- 5 Moallemi, E. A. *et al.* Early systems change necessary for catalyzing long-term sustainability in a post-2030 agenda. *One Earth* **5**, 792-811, doi:10.1016/j.oneear.2022.06.003 (2022).
- 6 GBD 2019 Risk Factors Collaborators. Global burden of 87 risk factors in 204 countries and territories, 1990–2019: a systematic analysis for the Global Burden of Disease Study 2019. *The Lancet* **396**, 1223-1249, doi:[https://doi.org/10.1016/S0140-6736\(20\)30752-2](https://doi.org/10.1016/S0140-6736(20)30752-2) (2020).
- 7 Li, K. *et al.* A two-pollutant strategy for improving ozone and particulate air quality in China. *Nature Geoscience* **12**, 906-910, doi:10.1038/s41561-019-0464-x (2019).
- 8 Patel, D. *et al.* Joint effects of heatwaves and air quality on ambulance services for vulnerable populations in Perth, western Australia. *Environmental Pollution* **252**, 532-542, doi:<https://doi.org/10.1016/j.envpol.2019.05.125> (2019).
- 9 Cohen, A. *et al.* Estimates and 25-year trends of the global burden of disease attributable to ambient air pollution: an analysis of data from the Global Burden of Diseases Study 2015. *The Lancet* **389**, 1907-1918, doi:10.1016/s0140-6736(17)30505-6 (2017).
- 10 Burnett, R. & Cohen, A. Relative Risk Functions for Estimating Excess Mortality Attributable to Outdoor PM_{2.5} Air Pollution: Evolution and State-of-the-Art. *Atmosphere* **11**, 589, doi:10.3390/atmos11060589 (2020).
- 11 Shen, H. *et al.* Urbanization-induced population migration has reduced ambient PM_{2.5} concentrations in China. *Science Advances* **3**, doi:10.1126/sciadv.1700300 (2017).
- 12 Liu, Y. *et al.* Role of climate goals and clean-air policies on reducing future air pollution deaths in China: a modelling study. *The Lancet Planetary Health* **6**, e92-e99, doi:10.1016/s2542-5196(21)00326-0 (2022).
- 13 Wang, Y. *et al.* Health Burden and economic impacts attributed to PM_{2.5} and O₃ in china from 2010 to 2050 under different representative concentration pathway scenarios. *Resources, Conservation and Recycling* **173**, 105731, doi:10.1016/j.resconrec.2021.105731 (2021).
- 14 Conibear, L. *et al.* The contribution of emission sources to the future air pollution disease burden in China. *Environmental Research Letters* **17**, 064027, doi:10.1088/1748-9326/ac6f6f (2022).
- 15 Kloog, I., Ridgway, B., Koutrakis, P., Coull, B. A. & Schwartz, J. D. Long- and Short-Term Exposure to PM_{2.5} and Mortality: Using Novel Exposure Models. *Epidemiology* **24** (2013).
- 16 Liu, C. *et al.* Ambient Particulate Air Pollution and Daily Mortality in 652 Cities. *New England Journal of Medicine* **381**, 705-715, doi:10.1056/NEJMoa1817364 (2019).
- 17 Silva, R. *et al.* Future global mortality from changes in air pollution attributable to climate change. *Nature Climate Change* **7**, 647-651, doi:10.1038/nclimate3354 (2017).
- 18 Yang, H., Huang, X., Westervelt, D. M., Horowitz, L. & Peng, W. Socio-demographic factors shaping the future global health burden from air pollution. *Nature Sustainability*, doi:10.1038/s41893-022-00976-8 (2022).
- 19 Hess, J. J. *et al.* Guidelines for Modeling and Reporting Health Effects of Climate Change Mitigation Actions. *Environmental Health Perspectives* **128**, 115001, doi:10.1289/EHP6745 (2020).
- 20 Huang, X., Srikrishnan, V., Lamontagne, J., Keller, K. & Peng, W. Effects of global climate mitigation on regional air quality and health. *Nature Sustainability*, doi:10.1038/s41893-023-01133-5 (2023).
- 21 Cheng, J. *et al.* A synergistic approach to air pollution control and carbon neutrality in China can avoid millions of premature deaths annually by 2060. *One Earth* **6**, 978-989, doi:10.1016/j.oneear.2023.07.007 (2023).
- 22 Girosi, F. & King, G. *Demographic Forecasting*, <<https://gking.harvard.edu/node/5502>> (2008).

- 23 Foreman, K. J. *et al.* Forecasting life expectancy, years of life lost, and all-cause and cause-specific mortality for 250 causes of death: reference and alternative scenarios for 2016-40 for 195 countries and territories. *The Lancet* **392**, 2052-2090, doi:10.1016/s0140-6736(18)31694-5 (2018).
- 24 Dellink, R., Chateau, J., Lanzi, E. & Magné, B. Long-term economic growth projections in the Shared Socioeconomic Pathways. *Global Environmental Change* **42**, 200-214, doi:10.1016/j.gloenvcha.2015.06.004 (2017).
- 25 Lutz, W., Anne, G., Samir, K., Marcin, S. & Nikolaos, S. *Demographic and Human Capital Scenarios for the 21st Century: 2018 assessment for 201 countries*, <<http://www.who.int/phe/publications/air-pollution-global-assessment/en/>> (2018).
- 26 Cifuentes, L., Borja-Aburto, V. H., Gouveia, N., Thurston, G. & Davis, D. L. Climate change: Hidden health benefits of greenhouse gas mitigation. *Science* **293**, 1257-1259, doi:10.1126/science.1063357 (2001).
- 27 Sandalow, D. *Guide to Chinese Climate Policy 2019*, <<https://energypolicy.columbia.edu/research/report/guide-chinese-climate-policy>> (2019).
- 28 Qian, H. *et al.* Air pollution reduction and climate co-benefits in China's industries. *Nature Sustainability*, doi:10.1038/s41893-020-00669-0 (2021).
- 29 Soergel, B. *et al.* A sustainable development pathway for climate action within the UN 2030 Agenda. *Nature Climate Change* **11**, 656-664, doi:10.1038/s41558-021-01098-3 (2021).
- 30 The State Council of China. *Fighting the Tough Battle of Pollution Prevention and Control*, <http://www.gov.cn/zhengce/2021-11/07/content_5649656.htm> (2021).
- 31 O'Neill, B. C. *et al.* The Scenario Model Intercomparison Project (ScenarioMIP) for CMIP6. *Geoscientific Model Development* **9**, 3461-3482, doi:10.5194/gmd-9-3461-2016 (2016).
- 32 Eyring, V. *et al.* Overview of the Coupled Model Intercomparison Project Phase 6 (CMIP6) experimental design and organization. *Geoscientific Model Development* **9**, 1937-1958, doi:10.5194/gmd-9-1937-2016 (2016).
- 33 Riahi, K. *et al.* The Shared Socioeconomic Pathways and their energy, land use, and greenhouse gas emissions implications: An overview. *Global Environmental Change* **42**, 153-168, doi:<https://doi.org/10.1016/j.gloenvcha.2016.05.009> (2017).
- 34 O'Neill, B. C. *et al.* Achievements and needs for the climate change scenario framework. *Nature Climate Change* **10**, 1074-1084, doi:10.1038/s41558-020-00952-0 (2020).
- 35 van Vuuren, D. P. *et al.* The representative concentration pathways: an overview. *Climatic Change* **109**, 5, doi:10.1007/s10584-011-0148-z (2011).

REVIEWER COMMENTS

Reviewer #2 (Remarks to the Author):

All comments have been addressed adequately and limitations acknowledged whenever possible. Thank you.

Reviewer #3 (Remarks to the Author):

The authors have addressed the issues raised in my previous review and provided detailed answers to my comments. In my opinion, the new re-framing of the study, and the additional clarifications and modifications responding to all reviewers have largely improved the clarity of the manuscript. However, I still have some comments.

First, I believe that the first subsection of the results ("Trends in DAPP under different scenarios and the attainment of SDG3.9.") is not a very relevant contribution, considering the large amount of literature exploring those health co-benefits for different scenarios (now revised and updated by the authors in the revised manuscript, Table S1). I still think that focusing more on the country-level (or spatial) outcomes along with the decomposition analysis (subsections #2 and #3 of the results) would make the manuscript more relevant for the community. Likewise, I think that the additional "sensitivity scenarios", lowering PM2.5 concentrations and improving baseline mortality rates by 20%, produce some expected results, and I do not see the value added of that exercise.

In addition, there are some dynamics that are confusing for me. For example:

"Global DAPP will not be substantially reduced under all but the most ambitious scenarios (SSP1-2.6) because of the overwhelming effect of population aging (Figure 3). However, there are complex interactions between different scenarios. For instance, with socioeconomic development, the population will get older which increases the size of vulnerable populations, but the death rate of diseases declines which simultaneously reduces population vulnerability. Hence, the effect of population aging will be offset by improvement in healthcare, hence scenarios with high population aging (SSP1-2.6 and SSP5-8.5) showed greater reduction in DAPP"

I believe that the scenario design (i.e., the combination of the scenarios with the 20% reduction of baseline mortality rates) can drive to these counterintuitive results, that would need to be revised in my opinion.

In summary, I think that this study provides relevant information and would be a contribution, while I believe that setting the focus at the national level, including the decomposition analysis of the different driving factors, would be more interesting for the research community.

Reviewer #4 (Remarks to the Author):

As far as I can see, the authors have substantially revised the manuscript and the supplement. Since I was specifically asked to assess whether the concerns raised in reviewer report 1 have been satisfactorily addressed, I will focus only on the corresponding changes below.

The presentation of the results of the individual simulations in new tables and supplementary figures addresses issue #1-1.

The question of whether an ensemble of models with partly oversimplified aerosol representations, which additionally requires an approximation to obtain consistent PM2.5 values, remains largely open.

Also, the aerosol particle size distribution is an important feature of each model, which is essentially modified by Eq. (3). However, despite these compromises, I believe that the ensemble strategy followed here is a valid and important approach that complements different approaches elsewhere, and I agree with the authors that judging the quality of individual models is beyond the scope of this study.

The impact of each factor on the achievement of SDG3.9 is now explained and discussed in detail. One point in this discussion relates to an aspect that seems to be missing in the "Limitations and Uncertainties" section. The saturation of the RR at high PM_{2.5} is much more pronounced in the MR-BRT model used here than in other risk functions, especially the Fusion model (Burnett et al. 2021, doi:10.1016/j.envres.2021.112245). The shortcomings of the MR-BRT, which led to the development of the GEMM and Fusion models, should at least be mentioned. In this context, a technical question: how are the age groups 15-20 and 20-25 mentioned in line 447 used when most MR-BRT functions are for ages above 25 years?

Overall, the concerns of reviewer 1 seem to be sufficiently addressed.

Detailed author response to reviewer comments: NCOMMS-23-06278A

Yue et al. "Substantially reducing global deaths from PM_{2.5} pollution under SDG3.9 requires advances in both air pollution control and healthcare"

We would like to express our gratitude to the editor and anonymous reviewers for their valuable comments and suggestions for improving the quality of the paper. We have carefully considered all the points raised by them. We are providing detailed point-by-point responses to all questions and recommendations by the reviewers. In the responses below, red fonts are the revised texts.

Response to Reviewer 2

Issue #2-1: All comments have been addressed adequately and limitations acknowledged whenever possible. Thank you.

Response #2-1: Thanks for your kind suggestions for improving our work.

Response to Reviewer 3

The authors have addressed the issues raised in my previous review and provided detailed answers to my comments. In my opinion, the new re-framing of the study, and the additional clarifications and modifications responding to all reviewers have largely improved the clarity of the manuscript. However, I still have some comments.

Issue #3-1: First, I believe that the first subsection of the results (“Trends in DAPP under different scenarios and the attainment of SDG3.9.”) is not a very relevant contribution, considering the large amount of literature exploring those health co-benefits for different scenarios (now revised and updated by the authors in the revised manuscript, Table S1). I still think that focusing more on the country-level (or spatial) outcomes along with the decomposition analysis (subsections #2 and #3 of the results) would make the manuscript more relevant for the community.

Response #3-1: Clarified and revised.

As the reviewer mentioned, we summarized previous studies that projected future trends of DAPP. Conflicting and inconsistent trends of DAPP among these studies is one main challenge for scholars to estimate the attainment of SDG3.9. Hence, one of the key contributions of this study is to estimate future trends in DAPP under a well-established scenario framework. Besides, the trends of DAPP are also the basis of further decomposition analysis at the country level. We restated our contribution in **Discussion**. The revised version is shown in line 307-316 as follows:

“Thus, our projections avoid unintended logical conflicts caused by using independent and internally inconsistent assumptions around driving factors¹ and improve the accuracy and robustness of projections.····· In addition, estimation based on a well-established scenario framework also makes our results more comparable with other studies under the SSPs and CMIP6 framework and estimates the trends in DAPP under different climate change and socioeconomic pathways. This can further support integrated decision-making by jointly considering the results from multiple fields^{2,3}.”

We appreciate the reviewer’s recognition that our results have the potential to support the community. We added some results of the decomposition analysis at the national level for representative countries like China and India which account for 50% of total DAPP worldwide in 2015 in the **Results** section. The revised version is shown in line 258-264 as follows:

“Detailed region- and country-specific decomposition were also conducted (Figure S19-20). For most countries, changes in aging and healthcare were the main contributors to DAPP but the interaction of other driving factors also had an influence. For instance, DAPP in China

significantly declined over time and met moderate or ambitious SDG3.9 targets under all scenarios except SSP3-7.0 because of low population growth and decreasing PM_{2.5} concentration. Conversely, DAPP in India tended to increase in most scenarios driven by population growth and relatively modest and slow air quality improvement.”

We have also provided national level results of the decomposition analysis in the supplementary material (Figure S20).

Fig. S1. Country-specific effects of different driving factors on the changes in DAPP from 2015 to 2030. Please refer to Error! Reference source not found. for detailed definitions of ASIA, OECD, MAF, REF and LAM.

Deeper decomposition analysis at the national level can be undertaken based on the publicly available dataset (<https://github.com/yuehuanbi/attainment-of-SDG3.9>) generated by this study for those with a specific interest in a deeper dive into the results than the space limits and constraints of this particular paper allows for.

Issue #3-2: Likewise, I think that the additional “sensitivity scenarios”, lowering PM_{2.5} concentrations and improving baseline mortality rates by 20%, produce some expected results, and I do not see the value added of that exercise. In addition, there are some dynamics that are confusing for me. For example: “Global DAPP will not be substantially reduced under all but the most ambitious scenarios (SSP1-2.6) because of the overwhelming effect of population aging (Figure 3). However, there are complex interactions between different scenarios. For instance, with socioeconomic development, the population will get older which increases the size of vulnerable populations, but the death rate of diseases declines which simultaneously reduces population vulnerability. Hence, the effect of population aging will be offset by improvement in healthcare, these scenarios with high population aging (SSP1-2.6 and SSP5-8.5) showed greater reduction in DAPP”. I believe that the scenario design (i.e., the combination of the scenarios with the 20% reduction of baseline mortality rates) can drive to these counterintuitive results, that would need to be revised in my opinion.

Response #3-2: Clarified and revised.

The additional sensitivity analysis aims to reveal how much effort needs to be invested for countries to meet SDG3.9 by adopting some relatively simple scenarios (lowering PM_{2.5} concentrations and improving baseline mortality rates by 20%) without considering the complex interactions among socioeconomic development, healthcare and aging (which is considered in the main results). While these simplifications ignore some effects of aging and lead to lower DAPP estimates, such additional efforts (lowering PM_{2.5} concentrations and improving baseline mortality rates by 20%) are more achievable via policymaking than lowering population aging. Therefore, we believe that it is useful to provide a basic insight into the non-linear response of DAPP to additional improvement in health care and air pollution control. We clarified this in the **Discussion** to reduce the potential for misunderstanding. The revised version is shown in line 434-443 as follows:

“When analyzing the additional improvement in health care and air pollution control, we used simple scenarios which generalize the complex interactions among socioeconomic development, healthcare, and aging. Policy efforts towards lowering PM_{2.5} concentrations and improving baseline mortality rates are far more achievable via policymaking than other measures for reducing DAPP such as lowering population aging and hence, these intervention

scenarios can provide general insights into the scale of response of DAPP required to meet SDG3.9. This simplification should be interpreted as a sensitivity analysis and may underestimate DAPP. More specific target setting and exploration of pathways by country considering emissions, energy structure, socioeconomic development, and political factors are needed^{4,5}.”

We also further clarified the driving mechanism of future DAPP under different scenarios. As we mentioned in the Table 1, the ScenarioMIP framework considering the complex interactions among socioeconomic development, healthcare and aging. For instance, high socioeconomic development will lead to better health care, and the people have higher life expectancy, embodied with stronger population aging. These interactions should be considered in interpreting the trend in DAPP. That’s why aging is the most influential driving factor that increases DAPP, but scenarios (especially SSP1-2.6 and SSP5-8.5) with highest aging have a relatively low DAPP in 2030. We clarified this in the **Discussion** to reduce the potential for misunderstanding. The revised version is shown in line 323-331 as follows:

“Global DAPP will not be substantially reduced under all but the most ambitious scenarios (SSP1-2.6) because of the overwhelming effect of population aging (**Error! Reference source not found.**). However, there are complex interactions between drivers under different scenarios which must be considered in terms of the net overall effect on DAPP. For instance, with socioeconomic development, the death rate of diseases declines reduces population vulnerability, but population aging simultaneously increases the size of vulnerable populations^{6,7}. Hence, while scenarios with high population aging (SSP1-2.6 and SSP5-8.5) counterintuitively showed a greater reduction in DAPP, this was driven by the effect of population aging being offset by improvement in healthcare in these scenarios.”

Issue #3-3: In summary, I think that this study provides relevant information and would be a contribution, while I believe that setting the focus at the national level, including the decomposition analysis of the different driving factors, would be more interesting for the research community.

Response #3-3: Clarified and revised.

Thanks for the reviewer’s recognition for our contribution. As we mentioned in **Response #3-1**, we now present some key results at the national scale. Deeper analysis at national scale to provide support for stakeholders and policymakers will be our next step. The revised version is shown in line 417-421 as follows:

“While we do not provide a detailed exploration of the results at the national-level here,

further analyses of trends and driving mechanism for individual countries may be undertaken based on the Supplementary Information provided to support decision-making of communities and stakeholders at regional and local scales⁸⁻¹⁰.”

Response to Reviewer 4

As far as I can see, the authors have substantially revised the manuscript and the supplement. Since I was specifically asked to assess whether the concerns raised in reviewer report 1 have been satisfactorily addressed, I will focus only on the corresponding changes below.

The presentation of the results of the individual simulations in new tables and supplementary figures addresses issue #1-1.

The question of whether an ensemble of models with partly oversimplified aerosol representations, which additionally requires an approximation to obtain consistent PM_{2.5} values, remains largely open. Also, the aerosol particle size distribution is an important feature of each model, which is essentially modified by Eq. (3). However, despite these compromises, I believe that the ensemble strategy followed here is a valid and important approach that complements different approaches elsewhere, and I agree with the authors that judging the quality of individual models is beyond the scope of this study.

The impact of each factor on the achievement of SDG3.9 is now explained and discussed in detail.

Issue #4-1: One point in this discussion relates to an aspect that seems to be missing in the "Limitations and Uncertainties" section. The saturation of the RR at high PM_{2.5} is much more pronounced in the MR-BRT model used here than in other risk functions, especially the Fusion model (Burnett et al. 2021, doi:10.1016/j.envres.2021.112245). The shortcomings of the MR-BRT, which led to the development of the GEMM and Fusion models, should at least be mentioned.

Response #4-1: Accepted and revised.

As the reviewer suggested, we clarified the potential shortcoming of the MR-BRT model in **Discussion**. The revised version is shown in line 407-414 as follows:

“Regarding the exposure-response function, we used the Bayesian, regularized, trimmed (MR-BRT) model following GBD2019 (see details in Methods), and presented the results derived from the middle value, and the upper and lower estimates was also conducted to quantify the range of uncertainty (Figure S7-8, Figure S22). Some studies also found a potential shortcoming of the MR-BRT model in that the saturation of relative risk at high PM_{2.5} tends to be more pronounced. More comprehensive sensitivity analyses considering other exposure-response functions (e.g., Global Exposure Mortality Model and Fusion model) are needed in the future to more thoroughly explore the influence of this modelling choice¹¹.”

Added references:

Burnett, R. T., *et al.* Designing health impact functions to assess marginal changes in outdoor

fine particulate matter. *Environmental Research* 204, 112245,
doi:<https://doi.org/10.1016/j.envres.2021.112245> (2022).

Issue #4-2: In this context, a technical question: how are the age groups 15-20 and 20-25 mentioned in line 447 used when most MR-BRT functions are for ages above 25 years? Overall, the concerns of reviewer 1 seem to be sufficiently addressed.

Response #4-2: Accepted and revised.

As the reviewer mentioned, we didn't consider the age groups 15-20 and 20-25 in our calculation. We revised the methods in line 477-478 as follows:

“In our formulation, 15 age groups were included in the equation, i.e., 25-30, 30-35, ..., 90-95, and 95+ years old.”

Reference:

- 1 Silva, R. *et al.* Future global mortality from changes in air pollution attributable to climate change. *Nature Climate Change* **7**, 647-651, doi:10.1038/nclimate3354 (2017).
- 2 Hess, J. J. *et al.* Guidelines for Modeling and Reporting Health Effects of Climate Change Mitigation Actions. *Environmental Health Perspectives* **128**, 115001, doi:10.1289/EHP6745 (2020).
- 3 Moallemi, E. A. *et al.* Early systems change necessary for catalyzing long-term sustainability in a post-2030 agenda. *One Earth* **5**, 792-811, doi:10.1016/j.oneear.2022.06.003 (2022).
- 4 Huang, X., Srikrishnan, V., Lamontagne, J., Keller, K. & Peng, W. Effects of global climate mitigation on regional air quality and health. *Nature Sustainability*, doi:10.1038/s41893-023-01133-5 (2023).
- 5 Cheng, J. *et al.* A synergistic approach to air pollution control and carbon neutrality in China can avoid millions of premature deaths annually by 2060. *One Earth* **6**, 978-989, doi:10.1016/j.oneear.2023.07.007 (2023).
- 6 Foreman, K. J. *et al.* Forecasting life expectancy, years of life lost, and all-cause and cause-specific mortality for 250 causes of death: reference and alternative scenarios for 2016-40 for 195 countries and territories. *The Lancet* **392**, 2052-2090, doi:10.1016/s0140-6736(18)31694-5 (2018).
- 7 KC, S. & Lutz, W. The human core of the shared socioeconomic pathways: Population scenarios by age, sex and level of education for all countries to 2100. *Global Environmental Change*, doi:10.1016/j.gloenvcha.2014.06.004 (2014).
- 8 Shen, H. *et al.* Urbanization-induced population migration has reduced ambient PM2.5 concentrations in China. *Science Advances* **3**, doi:10.1126/sciadv.1700300 (2017).
- 9 Liu, Y. *et al.* Role of climate goals and clean-air policies on reducing future air pollution deaths in China: a modelling study. *The Lancet Planetary Health* **6**, e92-e99, doi:10.1016/s2542-5196(21)00326-0 (2022).
- 10 Xu, F. *et al.* The challenge of population aging for mitigating deaths from PM2.5 air pollution in China. *Nature Communications* **14**, 5222, doi:10.1038/s41467-023-40908-4 (2023).
- 11 Burnett, R. T., Spadaro, J. V., Garcia, G. R. & Pope, C. A. Designing health impact functions to assess marginal changes in outdoor fine particulate matter. *Environmental Research* **204**, 112245, doi:<https://doi.org/10.1016/j.envres.2021.112245> (2022).

REVIEWERS' COMMENTS

Reviewer #3 (Remarks to the Author):

The authors have addressed the issues raised in my previous review and provided detailed answers to my comments.

I believe the new subnational analysis added to the main text (insights for China and India) and to the supplement (Figure S20) are really useful. Also, it is relevant for the community that the authors have included all the information in an open-access repository to carry out deeper national-level analyses.

The updated manuscript also clarifies some counterintuitive dynamics in the discussion section.

I have no additional comments or concerns, and I would recommend the study for publication.